# Insolation evolution and ice volume legacies determine interglacial and glacial intensity

Takahito Mitsui[1,2], Polychronis C. Tzedakis[3], Eric W. Wolff[4]

[1]Earth System Modelling, School of Engineering & Design, Technical University of Munich, Munich, Germany
[2]Potsdam Institute for Climate Impact Research, Potsdam, Germany
[3]Environmental Change Research Centre, Department of Geography, University College London, London, UK
[4]Department of Earth Sciences, University of Cambridge, Cambridge, UK

*Correspondence to*: Takahito Mitsui (takahito321@gmail.com)

**Abstract.** Interglacials and glacials represent low and high ice volume end-members of ice age cycles. While progress has been made in our understanding of how and when transitions between these states occur, their relative intensity has been lacking an explanatory framework. With a simple quantitative model, we show that over the last 800,000 years interglacial intensity can be described as a function of the strength of the previous glacial and the summer insolation at high latitudes in both hemispheres during the deglaciation. Since the precession components in the boreal and austral insolation counteract each other, the amplitude increase in obliquity cycles after 430,000 years ago is imprinted in interglacial intensities, contributing to the manifestation of the so-called Mid-Brunhes Event. Glacial intensity is also linked with the strength of the previous interglacial, the time elapsed from it, and the evolution of boreal summer insolation. Our results suggest that the memory of previous climate states and the time course of the insolation are crucial for understanding interglacial and glacial intensities.

## 1 Introduction

The most prominent climate signal in the last 800 thousand years (kyr) is the alternating pattern of glacials (with large ice sheets in high northern latitudes and an extended Antarctic Ice Sheet [AIS]), and interglacials (with little Northern Hemisphere [NH] ice outside Greenland [Past Interglacials Working Group of PAGES, 2016]) (Figs. 1d-g). It is generally accepted that this pattern is initiated and paced by changes in insolation caused by astronomical changes in orbit and axial tilt (Milankovitch, 1941) (Figs. 1a-c), modulated by strong feedbacks including those of the carbon cycle (Fig. 1d) and ice-sheet albedo. Both conceptual and sophisticated models have successfully reproduced many of the features of the observed record (e.g. Huybers, 2011; Parrenin and Paillard, 2012; Verbitsky et al., 2018; Berger et al., 1999; Abe-Ouchi et al., 2013; Willeit et al., 2019). However, it remains challenging to state simply what features of astronomical forcing determine the timing of glacial terminations, or the amplitude of glacial cycles. As a result, the holy grail of Quaternary palaeoclimate – to take only the external Milankovitch forcing and predict accurately the sequence and strength of glacial cycles, remains elusive. Recently a

simple rule (Tzedakis et al., 2017) was successful in predicting the timing of the occurrence of interglacials, but did not address their intensities.

Here, interglacial or glacial intensity (strength) refers to the magnitude of a measurable property that can be extracted from proxy climate records. Various metrics integrating global climate effects have been employed, including surface temperature

(Snyder, 2016; Past Interglacials Working Group, 2016), the oxygen isotope ratio ($\delta^{18}O$) in benthic Foraminifera in a stack of averaged globally-distributed records (Lisiecki and Raymo, 2005), sea-level reconstructions based on corals (Dutton et al., 2015), hydraulic control models of semi-isolated basins (Grant et al., 2014) and $\delta^{18}O$ of seawater (Elderfield et al., 2012). In terms of interglacial strength, the metrics converge in their broad trends over the last 800 kyr (Fig. 1): before the Mid-Brunhes Event (MBE; ~430 kyr before present [BP]), interglacials (Marine Isotope Stages [MIS] 19c, 17, 15e, 15a, 13a) were weaker

(cooler, higher benthic $\delta^{18}O$, atmospheric $CO_2$ lower than pre-industrial concentrations) (Berger and Wefer, 2003; Tzedakis et al., 2009). The strongest interglacials (MIS11c, 9e, 5e, 1) occurred after the MBE, although MIS7e and MIS7c are closer in intensity to pre-MBE interglacials. With respect to temperature and sea-level, MIS5e and MIS11c are the most prominent interglacials, followed by MIS9e and MIS1 (Elderfield et al., 2012; Grant et al., 2014; Dutton et al., 2015; Snyder et al., 2016; Past Interglacials Working Group of PAGES, 2016). The change in interglacial intensity at the MBE has been attributed to

different factors, including an increase in the amplitude of obliquity cycles (Fig. 1b) (EPICA Community Members 2004; Mitsui and Boers, 2021), increased instability of the West Antarctic Ice Sheet (Holden et al., 2011), changes in southern westerlies and Southern Ocean ventilation (Yin, 2013), and a reduction in the volume of interglacial Antarctic Bottom Water, contributing to a greater release of $CO_2$ from the deep ocean (Barth et al., 2018). Differences in the intensity of individual interglacials have been discussed in terms of the contribution of insolation and greenhouse gases (Yin and Berger, 2012; Obase

et al., 2021) and also in relation to ice-sheet size of the preceding glacial (Raymo, 1997; Paillard, 1998; Berger and Wefer, 2003; Lang and Wolff, 2010). However, a simple explanatory framework, providing a systematic understanding of these differences and trends has been lacking.

With respect to glacial strength, sea-level reconstructions (Elderfield et al., 2012; Grant et al., 2014; Shakun et al., 2015; Spratt

and Lisiecki, 2016) and glacial geologic evidence (Hughes and Gibbard, 2018; Batchelor et al., 2019; Hughes et al., 2020) indicate that the largest increases in global ice volume occurred during MIS12, 16, 2, 6, the smallest during MIS15b, 7d, 14, 8, with MIS10, 18, 20 somewhere in between. A decrease in boreal summer insolation is the primary trigger for glacial inception (Ganopolski et al., 2016), leading to rapid glacial advances in continental interiors and mountains (Hughes and Gibbard, 2018; Hughes et al., 2020). Within a glacial period, 'excess' ice volume accumulates during intervals of low

eccentricity-precession forcing leading to weak summer insolation maxima and minimum ice ablation (Raymo, 1997; Paillard, 1998). These observations underline the importance of the evolution of boreal insolation and time for ice-sheet buildup (Parrenin and Paillard, 2003) for differences in glacial strength.

In this work, we focus on the interglacial and glacial intensities over the last 800 kyr, where there is a broad consensus over which $\delta^{18}$O-peaks correspond to interglacials (Past Interglacials Working Group of PAGES, 2016). The extension of our models beyond 800 kyr BP will be the subject of a future study. As a stepping-stone to the more difficult target of predicting the amplitude of a long succession of glacial cycles from the forcing, here we address the question of predictability from one glacial trough to the next interglacial peak, and vice-versa over the last 800 kyr. If we know the astronomical forcing and some aspect of the preceding climate history, can we predict the strength of each interglacial and of each glacial? In anthropomorphic terms, could a bystander at a glacial maximum predict the strength of the interglacial that occurs about 20 kyr later, and could a bystander at the end of an interglacial predict the strength of the glacial maximum that might occur as long as 100 kyr later? We consider eleven interglacials over the last 800 kyr following Past Interglacials Working Group of PAGES (2016) and also take into account the interglacial MIS21 for predicting the glacial intensity in MIS20. We use the LR04 benthic $\delta^{18}$O stack as a reference record (Lisiecki and Raymo, 2005), on the premise that the convolved ice volume and deep-water temperature signal is a robust integrated metric of intensities (Past Interglacials Working Group of PAGES, 2016). Thus, the whole purpose of the present paper is to predict the amplitude of $\delta^{18}O$ from the insolation, the only external driver of the climate system. Atmospheric $CO_2$ or other climate feedbacks are considered as agents in-between insolation and $\delta^{18}O$ changes and their potential role is discussed in the final section.

## 2 Data and Methods

Given that uncertainties on the order of $\pm20$m in sea-level reconstructions from deconvolved records of deep-water temperatures and $\delta^{18}$O of seawater (e.g. Elderfield et al., 2012) exceed differences in interglacial highstands (Dutton et al., 2015), we use the LR04 benthic $\delta^{18}$O stack record (Lisiecki and Raymo, 2005) (mean standard error of 0.06‰) as an integrated metric of interglacial intensities. Although benthic $\delta^{18}$O records contain local deep-water temperature and hydrographic effects, averaging several globally-distributed records removes some of the regional variability. Interglacial intensities are measured by local minima in the LR04 stack at 2, 123, 217, 239, 329, 405, 491, 575, 610, 696, 780 and 858 kyr BP, respectively (Fig. 2a as well as 3a). Glacial intensities are measured by local maxima in the same record at 18, 140, 223, 252, 341, 433, 536, 585, 630, 718 and 794 kyr BP (Fig. 2a as well as 3a). According to Elderfield et al. (2012), during glacials deep-water temperatures rapidly decline and stabilize as they reach the ocean-water freezing temperature, resembling a square wave function; the saw-tooth character of the benthic $\delta^{18}$O records, therefore, reflects the slow ice-sheet buildup and rapid decay. By extension, differences in $\delta^{18}O_{max}$ between glacial maxima in the LR04 record largely reflect differences in ice volume. We assume that the orbital tuning in the LR04 $\delta^{18}$O stack record is essentially correct, at least on orbital time scales. Thus, we take it for granted which insolation peak induces which interglacial. Under this assumption, we explore the relationships between the amplitude (not the timing) of $\delta^{18}$O peaks and the insolation forcing (see Tzedakis et al. 2017 for the timing, which explains how one or two obliquity cycles are skipped without having terminations).

The caloric summer-half year insolation represents the amount of insolation integrated over the caloric summer half of the year, defined such that any day of the summer half receives more insolation than any day of the winter half (Milankovitch, 1941; Berger, 1978). Near $65°$N, the variance of this measure has almost equal contributions from climatic precession and the obliquity. Since ice ablation depends both on the strength of irradiation and the time period over which it is high (length of the summer season) (Milankovitch, 1941), we use this insolation metric in our models of interglacial and glacial intensities (Section 3). The caloric summer-half year insolation at $65°$N and at $65°$S is calculated at every 1 kyr by using R-package palinsol (Crucifix, 2016) based on the orbital solution of Laskar et al. (2004). Both insolations have almost the same average $\bar{F} = 5.845$ GJ m$^{-2}$ over the last 1 million years (Myr). The solar constant used is 1,368 Wm$^{-2}$.

In Section 3 we consider multiple linear regression models for interglacial intensities as well as for glacial intensities. We compare several candidate models using the Bayesian Information Criterion (BIC). The BIC is a common criterion for statistical model selection (Raftery, 1995) and is derived from the Bayes factor: the ratio of the marginal likelihoods of two competing models. Among different models that explain a dataset, the model with the lowest BIC is preferred. Any additional parameters in a model generally result in better or equally-good fits to the data, but can cause overfitting. Hence the BIC penalizes the increase in the number of parameters. When comparing two models (say model j against model k), the difference in BIC ($\Delta$BIC= BIC$_k$-BIC$_j$) is interpreted as the strength of the statistical evidence for model j against model k. As a rule of thumb (Raftery, 1995), the evidence is considered as *weak* (or *not worth more than a bare mention*) if 0<$\Delta$BIC<2, *positive* if 2<$\Delta$BIC<6, *strong* if 6<$\Delta$BIC<10, and *very strong* if $\Delta$BIC>10.

## 3 Results

### 3.1 Interglacial intensities

Figure 2a shows the LR04 benthic $\delta^{18}$O record over the last 800 kyr, where interglacial peaks ($\delta^{18}$O$_{\min}$) are indicated by red circles and glacial peaks ($\delta^{18}$O$_{\max}$) by blue triangles. In Fig. 2b, the caloric summer half-year insolation at $65°$N ($F_N(t)$, black) and that at $65°$S ($F_s(t)$, green) are shown respectively. The average of the two insolations ($\frac{1}{2}(F_N + F_S)$, magenta) yields a variation correlated to obliquity ($R = 0.998$) because the climatic precessions across the two hemispheres cancel each other (Milankovitch, 1941; Berger, 1978). Comparing Figs 2a and 2b, we observe that each termination starts near the time when the boreal summer insolation $F_N(t)$ exceeds its average $\bar{F} = 5.845$ GJ m$^{-2}$. Exceptionally, the start of Termination III leading to MIS7e is delayed relative to the crossing point between $F_N(t)$ and $\bar{F}$ by about 11 kyr, possibly because the rise in the insolation $F_N(t)$ stops (and slightly decreases) around a local minimum at 254 kyr BP. Termination III starts after the local minimum at 254 kyr BP, responding to the second rise in the insolation. We denote those upward crossing points between $F_N(t)$ and $\bar{F}$ associated with terminations by $t_s$ (including an exception $t_s = 254$ kyr BP for Termination III; the case without this exception is discussed below). We further observe that each termination is completed near the time ($t_e$) when the

insolation $F_N(t)$ falls below the average $\bar{F}$. While most of the deglaciation intervals contain one insolation maximum, the interval leading to MIS17 contains two insolation maxima, and the interval leading to MIS13a three insolation maxima.

We seek a simple regression relation describing interglacial intensity, represented by $\delta^{18}O_{min}$. We postulate that the magnitude of each termination is related to the boreal summer insolation anomaly integrated during the insolation-based termination period $[t_s, t_e]$ at 65°N and the austral summer insolation (at 65°S) anomaly integrated during the same period:

$$I_N = \sum_{t=t_s}^{t_e} (F_N(t) - \bar{F}), \qquad I_S = \sum_{t=t_s}^{t_e} (F_S(t) - \bar{F})$$

These quantities shown in Fig. 2c are used as explanatory variables for the interglacial intensities. In addition to these, we
consider the average, $I_{AV} = \frac{1}{2}(I_N+I_S)$, as an alternative explanatory variable. We then include the previous glacial intensity $\delta^{18}O_{max}$ as another explanatory variable (Fig. 2d), based on the observation that strong interglacials often follow strong glacials (Lang and Wolff, 2011). Figure 3 shows that these variables are able to separate the strongest interglacials from the rest. In the present framework, the timing and intensity of glacial maxima ($t_s$ and $\delta^{18}O_{max}$) are taken from the palaeodata, while the end of complete deglaciations, $t_e$, is known from a predictive model (Tzedakis et al., 2017) that indicates which
insolation cycles lead to complete deglaciations.

We perform a linear regression of $\delta^{18}O_{min}$ with equation $\delta^{18}O_{min} = \beta_0 + \beta_1 \delta^{18}O_{max} + \beta_2 I_N + \beta_3 I_S$. We refer to this case in which $\beta_2$ and $\beta_3$ are chosen independently as Model 2. Model 1 does not use the term in $I_S$ (i.e., $\beta_3 \equiv 0$). We also consider Model 3 defined by the regression equation $\delta^{18}O_{min} = \beta_0 + \beta_1 \delta^{18}O_{max} + \beta_2 I_{AV}$, which is a special case of Model 2 in which
the coefficients of $I_N$ and $I_S$ are equal. In all three models, the actual and predicted $\delta^{18}O_{min}$ are strongly correlated (Fig. 2e, Table 1). Model 2, which has an extra parameter, gives the highest correlation 0.94 (more variability explained), while model 3 using the average insolation integral $I_{AV}$ provides a similar level of correlation 0.92 as model 2. In all three models, all the explanatory variables are significant (Table 1). According to the BIC (Section 2.3), model 2 and model 3 have *strong* evidence against model 1 ($\Delta$BIC>8), indicating that insolations in both hemispheres are crucial for the differences in interglacial
intensities. On the other hand, the evidence of model 2 against model 3 is *weak* ($\Delta$BIC<2), that is, they are almost equally supported by the data.

Predictions of all the regression models are consistent with the observation that MIS11c, 9e, 5e, 1 are stronger than other interglacials over the last 800 kyr (Fig. 2e). Predictions of MIS15e and MIS11c diverge from observations in Model 1, but are
155 well reproduced in Models 2 and 3, which include $I_S$. We also tested simpler models without the $\delta^{18}O_{max}$ term, but they give poor fits, particularly for the strength of MIS7e, 7c and 5e compared to the original models (Fig. S1 and Table S1 in the Supplement). The contribution of $\delta^{18}O_{max}$ is also evident from the BIC (the BICs for the original models are substantially lower than those for the simpler models). In Appendix A, we also explored the effect of using the first crossing point between

$F_N(t)$ and $\bar{F}$ for the starting time $t_s$ of Termination III on our model results: the predicted interglacial intensity $\delta^{18}O_{min}$ of MIS7e is stronger than the observation (Fig. A1) and the prediction skills slightly decrease (Table A1), but the main features of interglacial intensities are reproduced in models 2 and 3. Taken together, our results indicate that interglacial intensity is determined by the strength of the preceding glacial and the summer energy received during deglaciation at both polar regions.

## 3.2 Glacial intensities

We now attempt to explain glacial intensity $\delta^{18}O_{max}$ on the premise that we know the timing of the glacial maximum $t_{max}$, the timing of the previous interglacial peak $t_{min}$ and its intensity $\delta^{18}O_{min}$ from observations.

Inspection of Fig. 4a shows that a longer glacial duration ($T = t_{max} - t_{min}$) typically results in a larger increase of $\delta^{18}O$ (larger $\delta^{18}O_{max} - \delta^{18}O_{min}$). For example, the two weakest glacials MIS15b and MIS7b are also the shortest, while strong glacials are longer ($T \gtrsim 60$ kyr) (Fig. 4e). However, the relation between $\delta^{18}O_{max} - \delta^{18}O_{min}$ and $T$ is nonlinear. Figure 5a shows the time evolution of $\delta^{18}O$ during each glacial period relative to the time elapsed since the previous interglacial peak $\delta^{18}O_{min}$. While the time evolution varies with each glacial, the rate of increase in $\delta^{18}O$ is typically higher in the beginning ($T \lesssim 20$ kyr) (reflecting in large part rapid deep-ocean cooling [Elderfield et al., 2012]) and decreases as time elapses (or as $\delta^{18}O$ increases). As a zeroth order approximation, the increase in $\delta^{18}O$ may be expressed by a linear relaxation process $\frac{dx}{dt} = (A - x)/\tau$, $x(0) = 0$. Its solution $x(t) = A(1 - e^{-t/\tau})$ roughly fits the observed $\delta^{18}O$ changes in Fig. 5a (magenta solid line) for $\tau \sim 25$ kyr and $A \sim 1.3$ ‰ (the same functional form is assumed for an ice sheet response in Imbrie et al., 1993). In the following model, $x(T)$ represents the basic time dependence of $\delta^{18}O$ change in a glacial period with duration $T$ (kyr).

We then consider the effect of insolation. As observed in Fig. 5b, the growth rate of $\delta^{18}O$ ($d\delta^{18}O/dt$) is high when the caloric summer insolation at 65°N is low, while the average growth rate is close to zero for high insolation values. Specifically $\delta^{18}O$ almost exclusively grows (at ~10-kyr time scales) for values of insolation below ~5.7—5.8 GJ m$^{-2}$. We therefore introduce a measure for low insolation periods (Fig. 4d): the total time $L$ (kyr) within a glacial period $[t_{min}, t_{max}]$ during which the caloric summer insolation at 65°N is below an empirical threshold 5.7 GJ m$^{-2}$ (see below for this specific value). We obtain a similar result if we use the integral of the insolation below a threshold (Fig. S2) instead of the total time, while the model fit using the total time $L$ is slightly better than this alternative. In either case, a measure of insolation deficit contributes to the prediction of glacial intensities. Thresholding is the simplest way to capture insolation deficits.

We assume that the $\delta^{18}O$ increase between an interglacial minimum and the ensuing glacial maximum ($\delta^{18}O_{min} - \delta^{18}O_{max}$) is decomposed into the basic time dependence $1 - e^{-\frac{T}{25}}$ and the total low-insolation period $L$ (kyr) during the period of glaciation (Fig. 4). That is, we suppose the relation $\delta^{18}O_{max} - \delta^{18}O_{min} = \beta_0 + \beta_1 \left(1 - e^{-\frac{T}{25}}\right) + \beta_2 L$. The result of the

regression analysis is given in Table 2. The estimated intercept $\beta_0$ is quite small $-0.033$ ‰ and the null hypothesis $\beta_0 = 0$ cannot be rejected ($p = 0.88$). If this model is compared with the one without intercept, i.e., $\beta_0 \equiv 0$ (Table 2), the BIC positively supports the latter ($\Delta$ BIC>2). Therefore, we select the model without intercept $\beta_0$, that is, $\delta^{18}O_{max} - \delta^{18}O_{min} = 1.4\left(1 - e^{-\frac{T}{25}}\right) + 0.025L$. The predicted $\delta^{18}O_{max}$, shown in Fig. 4g, is strongly correlated with observations (R=0.90). The p-values for each coefficient are both less than 0.005 suggesting that both explanatory variables are important. What emerges is that the length of the glacial between the interglacial peak and the glacial maximum, the total time during which insolation is below a threshold, and the $\delta^{18}O_{min}$ of the preceding interglacial all have a large impact on the intensity of the ensuing glacial maximum.

The final model has four parameters: regression coefficients $\beta_{1,2}$, time constant $\tau$ ($= 25$ kyr) and the threshold defining the low-insolation period (5.7 GJ m$^{-2}$). The latter two values are suggested from the observations (Figs. 4 and 5) and are adopted because they provide a good prediction. The result is however relatively insensitive to those parameters; we obtain a good prediction with correlation coefficient close to 0.9 in a range of $\tau$ and threshold values, as shown in Fig. S3. The model with four parameters might be considered complex for predicting 11 glacial intensities. If the model is severely overfitted to the data, the model would not possess prediction ability for new data. In Appendix B, we have shown by the leave-one-out cross validation method that the model actually has prediction ability for unseen data. It should also be noted that in this model, the glacial duration $T$ is taken from the LR04 benthic $\delta^{18}O$ record assuming that its orbital tuning is right.

**4 Summary and Discussion**

We have inspected the predictability of interglacial and glacial intensities over the last 800 kyr. Interglacial strength can be well-predicted at the previous glacial maximum with the caveat that in two cases, the amplitude was achieved only in the second or third insolation peak. Glacial strength is well-predicted at the previous interglacial peak, with the caveat that the length of the glacial is currently taken from the data. While the models contain three to four parameters, they are still simple explanatory frameworks. These models show that interglacial intensity over the last 800-kyr can be described as a function of the strength of the previous glacial and the summer insolation at high latitudes in both hemispheres during the deglaciation, and that glacial intensity is linked with the strength of the previous interglacial, the time elapsed from it, and the evolution of boreal summer insolation.

While previous studies (e.g. Lang and Wolff, 2011) have underlined the influence of large ice sheets on the extent of deglaciation, our analysis provides a quantitative description of its contribution. Figure 6a shows the decomposition of the predicted $\delta^{18}O_{min}$ anomaly into the contribution of different factors: the weak MIS15e deglaciation is attributed to the low average insolation integral $I_{AV}$, even though it follows one of the strongest glaciations, MIS16; on the other hand, the weak

deglaciations of MIS15a, 13a, 7c, 7e are attributed to the weak MIS15b, 14, 7d, 8 glacials, respectively. Large ice sheets are more unstable as a result of ice-sheet physics, glacio-isostatic adjustments, extensions to lower latitude, and changes in ice-sheet albedo (MacAyeal, 1979; Birchfield et al., 1981; Marshall and Clark, 2002; Ganopolski et al., 2010; Abe-Ouchi et al., 2013) and are therefore more sensitive to insolation increases. In addition, the effect of glacial intensity on deglaciation might be operating partly through its influence on atmospheric $CO_2$. Larger NH ice sheets have a capacity to produce larger amounts of freshwater or a longer period of freshwater discharge into the North Atlantic, weakening the Atlantic Meridional Overturning Circulation and leading to activation of the bipolar-seesaw (Knutti et al., 2004; Denton et al., 2010; Tzedakis et al., 2022). Large ice sheets may also promote stronger deep-ocean salinity stratification, which stabilizes relatively warm waters in the glacial deep ocean and then amplifies the rate of Antarctic warming during the activation of the bipolar seesaw of ensuing deglaciation (Knorr et al., 2021). These changes in turn lead to more $CO_2$ outgassing from the Southern Ocean (Watson and Garabato 2006; Stephens and Keeling, 2000), accelerating ice-sheet ablation and are consistent with the combined role of insolation and $CO_2$ proposed by Yin and Berger (2012).

Our analysis also developed an empirical relation linking glacial intensity with the length of the preceding glacial period, the total length of low insolation periods during that glacial, and the preceding interglacial intensity. The decomposition of the predicted $\delta^{18}O_{max}$ anomaly into the contributions from these factors (Fig. 6b) shows that all pre-MBE glacials (and also MIS7d and 6) were strengthened by the extent of the $\delta^{18}O_{min}$. This represents remnant ice at the end of the preceding interglacial that contributes by reducing albedo and as a seed for ice growth that occurs during periods of low insolation. The decomposition also reveals that MIS12, the largest Quaternary glaciation, is the only occasion when all three factors make a positive contribution.

**The MBE**. A striking feature of our results is the change in the time integrals of the averaged boreal and austral insolation, $I_{AV,}$ across the MBE (Fig. 2c). The decomposition of the predicted $\delta^{18}O_{min}$ anomaly (Fig. 6a) clearly shows that while $\delta^{18}O_{max}$ has positive contributions before and after the MBE, a shift from negative to positive contributions is observed in $I_{AV}$ after the MBE. As previously discussed, $I_{AV}$ is dominated by obliquity, which has a strong influence on the insolation received over the local summer season and over the year at high latitudes of both hemispheres (Milankovitch, 1941). The influence of obliquity on high latitudes is further amplified by sea-ice and snow-albedo feedbacks, while the contribution of precession may be stronger at northern high latitudes (Yin and Berger, 2012). The EPICA Dome C sea-salt Na flux record, a proxy for sea-ice extent (Wolff et al., 2006), shows a reduction in interglacial sea-ice after the MBE (Fig. 1e), which may have allowed more $CO_2$ outgassing from the Southern Ocean (Watson and Garabato 2006; Stephens and Keeling, 2000), leading to increased global annual mean temperatures and ice ablation. In addition, the stronger post-MBE component of $I_{AV}$ and also $I_S$ during deglaciations, enhancing the melting of sea-ice and warming of the Southern Ocean, may have destabilized the AIS increasing its contribution to sea-level highstands. This is consistent with offshore sediment and geochemical data on provenance changes, suggesting increased ice loss from the Wilkes Subglacial Basin during post-MBE interglacials (MIS11c, 5e, 9e), although the

pre-MBE record remains weakly constrained (Wilson et al., 2021). Recently Mitsui and Boers (2021) have developed an Artificial Neural Network (ANN) model that performs a skilful 21-kyr ahead prediction of $\delta^{18}O$ on the basis of the past $\delta^{18}O$ history and the insolation evolution. Through the sensitivity analysis of the ANN model, they concluded that the intensification of interglacials across the MBE is attributed to the amplitude increase in the obliquity forcing. While this is consistent with our conclusions, our present regression model is more physically interpretable than the ANN model and even more precise in predicting $\delta^{18}O_{min}$.

Thus, the shift in interglacial intensities at the MBE may be ultimately related to the amplitude-modulation of obliquity with a duration of ~1.2 Myr (Lourens and Hilgen, 1997), which led to higher obliquity variations after 430 kyr BP. If this conjecture is correct, then a similar shift from weaker to stronger interglacials should have occurred about 1.6 Myr BP. The LR04 stack (Fig. 7) hints at an interval of weaker interglacials ~1.9-1.6 Myr BP, but the averaging of several records to create the stack tends to smooth $\delta^{18}O$ variability. By comparison, inspection of the Shackleton05 composite benthic $\delta^{18}O$ record from the Eastern Equatorial Pacific (see Tzedakis et al., 2017 for references) shows more clearly the occurrence of weaker interglacials from around 1.9 to 1.6 Myr BP and a shift to stronger interglacials after that. Although it would be interesting to explore whether our model can reproduce a shift in interglacial intensities at ~1.6 Myr BP, uncertainties over the size of ice sheets and the deep-water temperature components of $\delta^{18}O$ complicate such an undertaking at present.

While we remain some distance from a fully predictive model of temperature, ice volume and sea level over the entire sequence of glacial cycles, our analysis lays out some of the key predictors that need to be understood in physical models and coupled together, and underlines the importance of ice volume legacies and the time course of insolation on the amplitude of glacial cycles.

**Appendix A: Sensitivity on the starting time of the Termination**

In the models of interglacial intensities in Section 3.1, the starting time $t_s$ of Termination is chosen as the first crossing point between $F_N(t)$ and $\bar{F}$ except for Termination III (Fig. 2). Termination III started after the local insolation minimum at 254 kyr BP (orange dotted line) behind the crossing point by about 11 kyr, responding to the second rise in the insolation. This is why we set $t_s = 254$ kyr BP for Termination III in our main models in Fig. 2. Here we show the effect of using the first crossing point between $F_N(t)$ and $\bar{F}$ for the starting time of Termination III on our model results: the predicted interglacial intensity $\delta^{18}O_{min}$ of MIS7e is stronger than the observation (Fig. A1) and the prediction skills slightly decrease (Table A1), but the main features of interglacial intensities are reproduced in models 2 and 3.

**Appendix B: Prediction ability of the models for unseen data**

Models that contain more unknown parameters than can be justified by the data are called overfitted (Everitt and Skrondal, 2010). Overfitted models can have low prediction ability for new data, even if they appear skilful for training data. The leave-one-out cross validation (LOOCV) approach is one of the methods to assess the prediction ability of a machine learning model for new data: A model is trained with N-1 data points removing 1 data point from the entire data set with N points (here N=11). The trained model is then used to predict the removed data point. This procedure is applied for every data point. Based on the

average prediction error in LOCCV, we can infer the prediction ability of the model for unseen data. If a regression model is severely overfitted to data, the correlation coefficient between the data and the prediction in LOCCV would substantially decrease from the correlation coefficient obtained by usual model fitting.

Model 1 for the interglacial intensities gives a correlation coefficient of R=0.70 in LOOCV, while the correlation coefficient

in the usual fitting is R=0.83 (Table 1). Model 2 for the interglacial intensities gives R=0.82 in LOOCV and R=0.94 in the usual fitting (Table 1). Model 3 for the interglacial intensities gives R=0.85 in LOOCV and R=0.92 in the usual fitting   (Table 1). Model 2 for glacial intensities gives R=0.84 in LOOCV and R=0.90 in the usual fitting (Table 2). In sum, the correlation in LOOCV is slightly lower than the correlation in usual fitting, but the difference is not substantially large in each model. That is, our models with three to four parameters have prediction ability for unseen data, even if trained with 10 data points.

Therefore we consider that the models are not severely overfitted. Although the number of parameters (3-4) is large in comparison with the number of data points N=11, the models are reasonably simple compared to the complexity of ice age cycles arising from various feedbacks.

**Code and data availability**

R-package palinsol version 0.97 (Crucifix, 2016) used for calculating the caloric summer-half year insolation at $65°$N and at

305 $65°$S is available from https://cran.r-project.org/web/packages/palinsol/index.html. The postprocessed data used for Figs. 2 and 4 are provided as supplementary materials. The R-codes used in this study are available from the corresponding author (Takahito Mitsui) upon reasonable request.

**Supplement**

The postprocessed data used for Figs. 2 and 4 are provided as supplementary materials.

**Author contributions**

PCT and TM conceived the study. TM conducted the analyses with contributions from PCT and EWW. All three authors interpreted and discussed results and wrote the manuscript.

**Competing interests**

EWW is a member of the editorial board of Climate of the Past. The authors have also no other competing interests to declare.

**Acknowledgements**

The authors thank Michel Crucifix who modified the R-package palinsol in order to compute the Southern Hemisphere caloric summer half-year insolation. TM acknowledges funding by the Volkswagen Foundation. PCT acknowledges funding from the UK Natural Environment Research Council (NE/V001620/1). EWW is supported by a Royal Society Professorship. This is a contribution to the PAGES Working Group on Quaternary Interglacials (QUIGS).

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

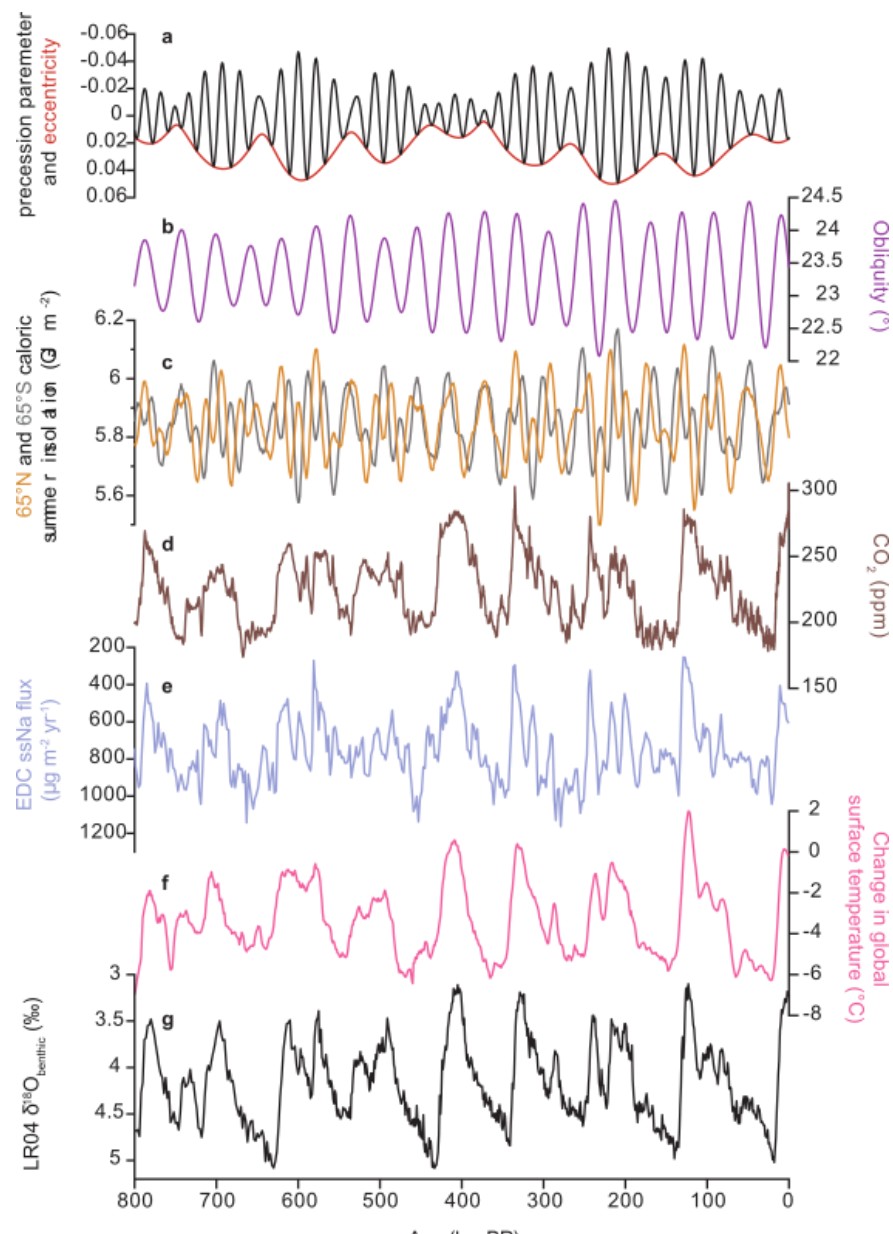

**Figure 1: Changes in climate conditions over the last 800 kyr. (a) Eccentricity (red) and precession parameter (black) (Laskar et al., 2004). (b) Obliquity (Laskar et al., 2004). (c) Caloric summer half-year insolation at 65°N (orange) and at 65°S (grey) based on the orbital solution of Laskar et al., 2004. (d) Compilation of atmospheric $CO_2$ records from Antarctic ice cores (Bereiter et al., 2015, and references therein; Nehrbass-Ahles et al., 2020; Shin et al., 2020; Bauska et al., 2021). (e) EPICA Dome C sea salt Na flux, a proxy for sea-ice extent (Wolff et al., 2006). (f) Global average surface temperature as temperature deviation from present (average over 0-5 kyr BP) (Snyder, 2016). (g) Stack of benthic δ18O records, LR04 (Lisiecki and Raymo, 2005). Ice core data are on the AICC2012 timescale (Bazin et al., 2013).**

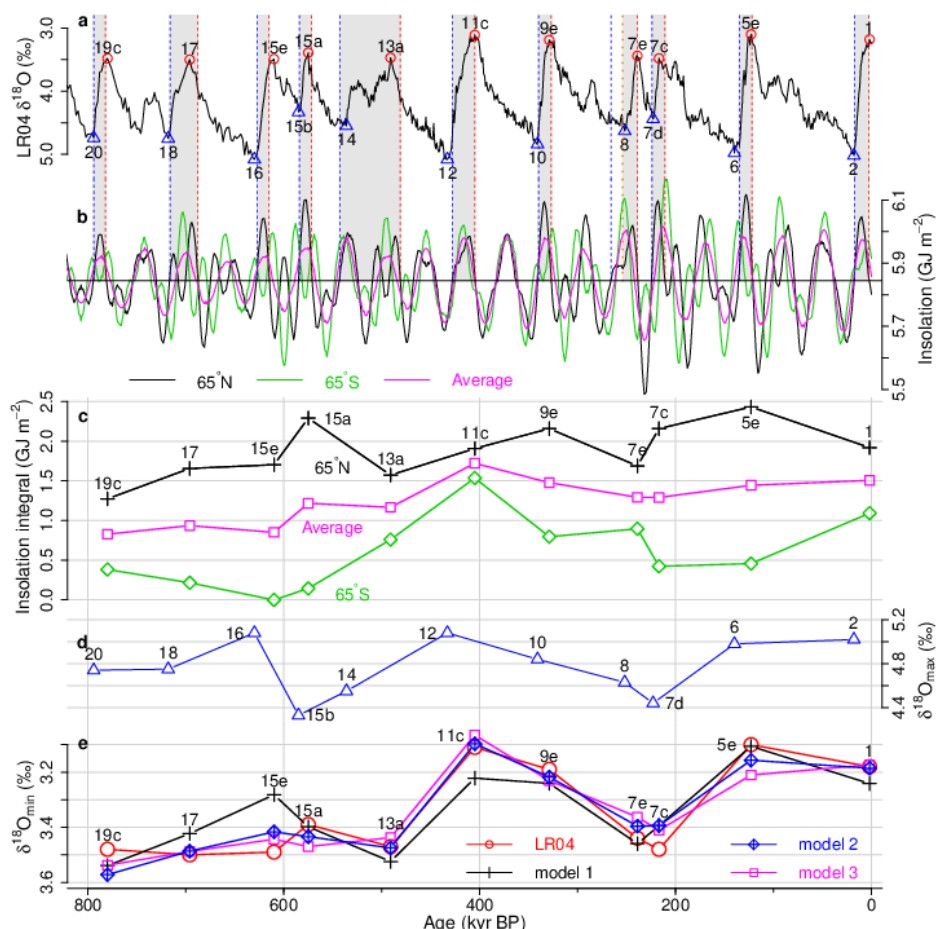

**Figure 2: Modelling interglacial intensities. (a)** LR04 $\delta^{18}O$. The red circles indicate the minima of $\delta^{18}O$ ($\delta^{18}O_{min}$) at each interglacial, and the blue triangles the maxima ($\delta^{18}O_{max}$) at glacials. See below for the grey strips and the dashed lines. **(b)** Caloric summer half-year insolation at 65°N ($F_N$, black) and 65°S ($F_S$, green). The average of the two (magenta) is also shown. The blue dashed lines show timings $t_s$ at which the caloric summer half-year insolation at 65°N exceeds average 5.845 GJ m$^{-2}$ (black horizontal line) and the red dashed lines show timings $t_e$ at which the insolation falls back below the average. Each termination starts roughly around $t_s$, and it is completed around $t_e$. Exceptionally termination III starts after the local insolation minimum at 254 kyr BP (orange dotted line), responding to the second rise in the insolation. The grey strips show the termination intervals $[t_s, t_e]$ based on the insolation curve. **(c)** Integral of the caloric summer insolation anomaly between $t_s$ and $t_e$ at 65°N, $I_N$ (black cross), the integral at 65°S for the same period, $I_S$ (green diamond), and the average $I_{AV} = \frac{1}{2}(I_N + I_S)$ (magenta square). **(d)** $\delta^{18}O_{max}$. **(e)** Predictions by linear regression models with explanatory variables in (c) and (d): Model 1 with $I_N$ (black cross); model 2 with both $I_N$ and $I_S$ with their own coefficients (blue diamond with cross); model 3 with $I_{AV}$ (magenta squares).

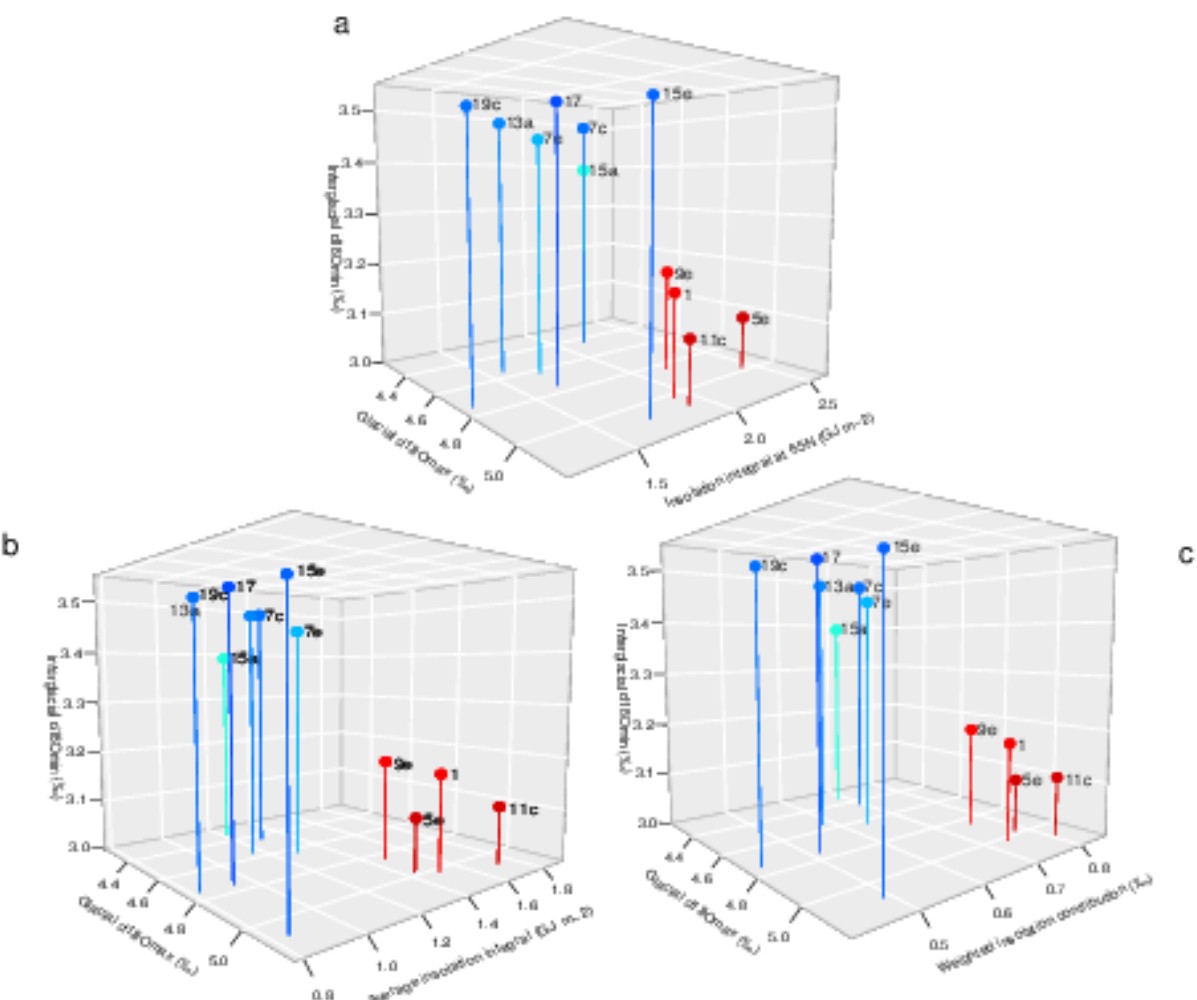

**Figure 3: Scatter plot in the space: (a) ($\delta^{18}O_{max}$, $I_N$, $\delta^{18}O_{min}$), (b) ($\delta^{18}O_{max}$, $I_{AV}$, $\delta^{18}O_{min}$) and (c) ($\delta^{18}O_{max}$, $\beta_2 I_N + \beta_3 I_S$,**
**$\delta^{18}O_{min}$). (a) We can separate the four strong interglacials (MIS11c, 9e, 5e, and 1) from the weak interglacials in the plane of explanatory variables ($\delta^{18}O_{max}$, $I_N$). In this case, however, weak interglacial MIS15e is close to the strong interglacials in $\delta^{18}O_{max}-I_N$ plane. Replacing $I_N$ with a mixture of $I_N$ and $I_S$ (b, c) leads to a clear separation of the four strongest interglacials (MIS11c, 9e, 5e, 1) from the rest.**

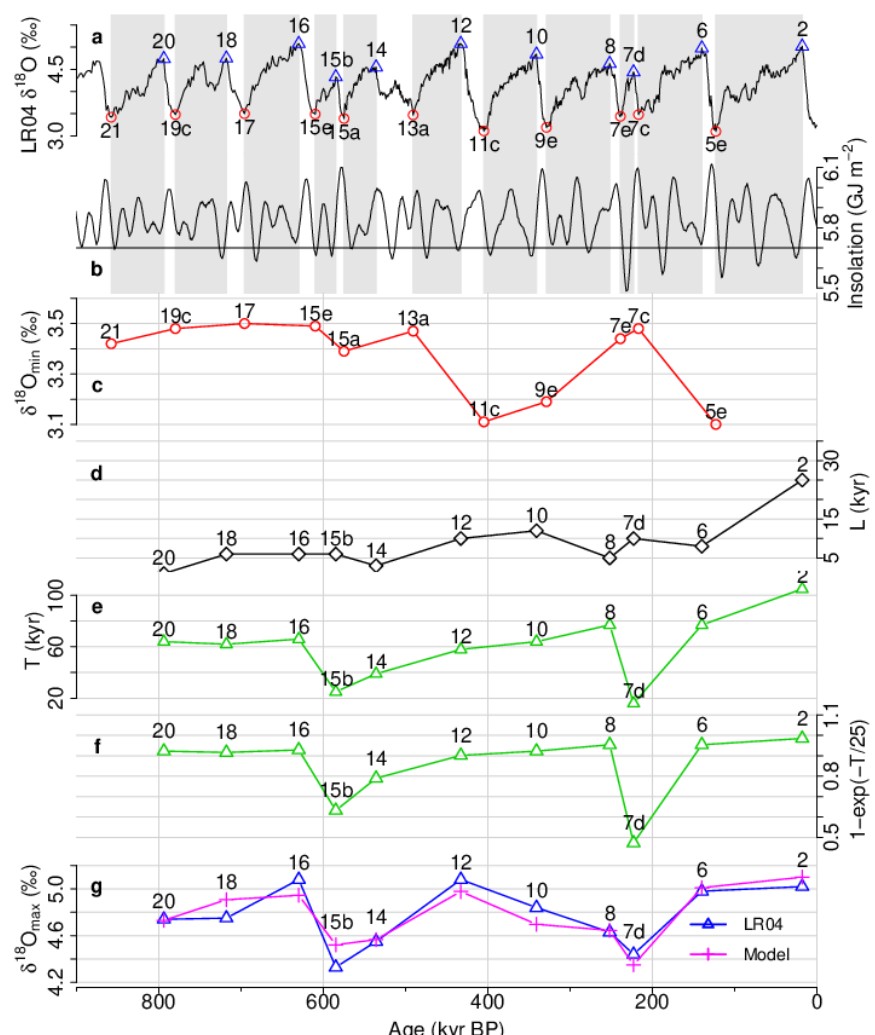


**Figure 4: Modelling glacial intensities. (a)** LR04 $\delta^{18}$O. The red circles indicate $\delta^{18}$O$_{min}$ at each interglacial, and the blue triangles the maxima $\delta^{18}$O$_{min}$ at glacials. The time intervals between them are shaded. Note that the data are plotted inversely to Fig. 2, with glacial maxima above interglacial minima. **(b)** Caloric summer insolation at 65°N. The grey shading is the same as in (a). **(c)** $\delta^{18}$O$_{min}$ for each interglacial. **(d)** Total time $L$ during which the caloric summer insolation is below a threshold $5.7$ GJ m$^{-2}$ between the interglacial peak and the glacial peak. **(e)** Time span $T$ between the interglacial peak and the glacial peak. **(f)** $1 - e^{-T/25}$. **(g)** Prediction of $\delta^{18}$O$_{max}$ from the linear regression relation with explanatory variables in (d) and (f) (R=0.90).


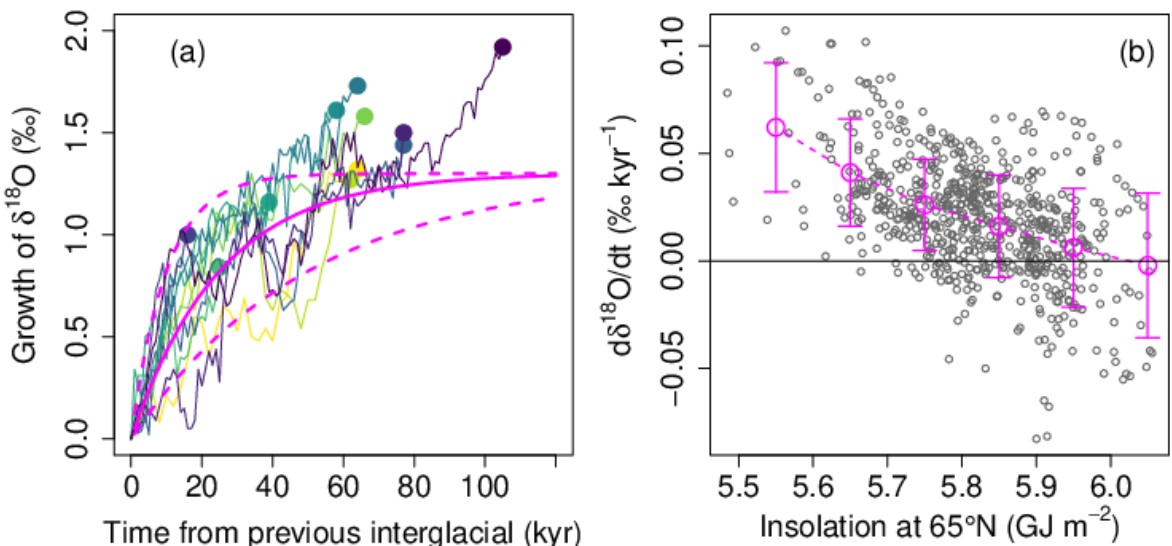

**Figure 5: (a)** Time evolution of $\delta^{18}O$ during each glacial period from its previous interglacial $\delta^{18}O_{min}$. The magenta lines are baseline increase profiles $x(t) = A(1 - e^{-t/\tau})$ for $\tau = 10, 25,$ and $50$ kyr and $A = 1.3$ ‰. **(b)** Growth rate of $\delta^{18}O$ ($d\delta^{18}O/dt$) vs the caloric summer insolation at 65°N only during a glacial period $[t_{min}, t_{max}]$. $d\delta^{18}O/dt$ is calculated after 15-point Gaussian smoothing on 1 kyr re-sampled data. The magenta circle with 1-sigma range shows the mean in a bin of the caloric summer insolation of size 0.1 GJ m$^{-2}$.


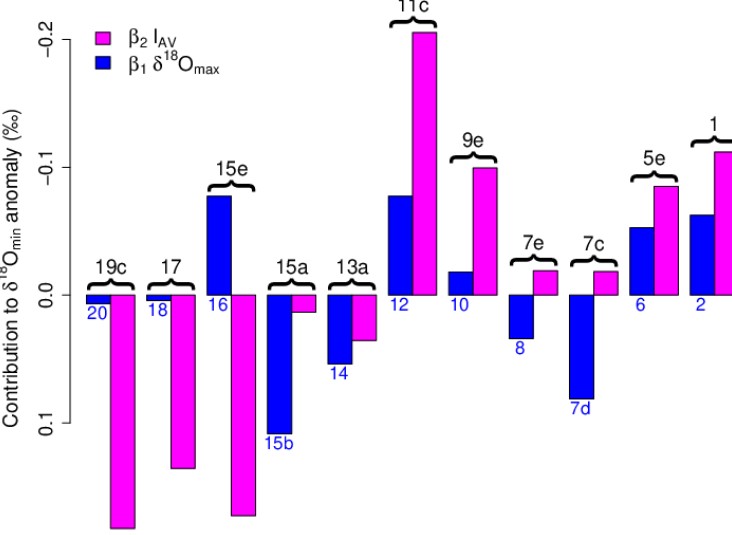

**(a) Model 3 for interglacial intensities**

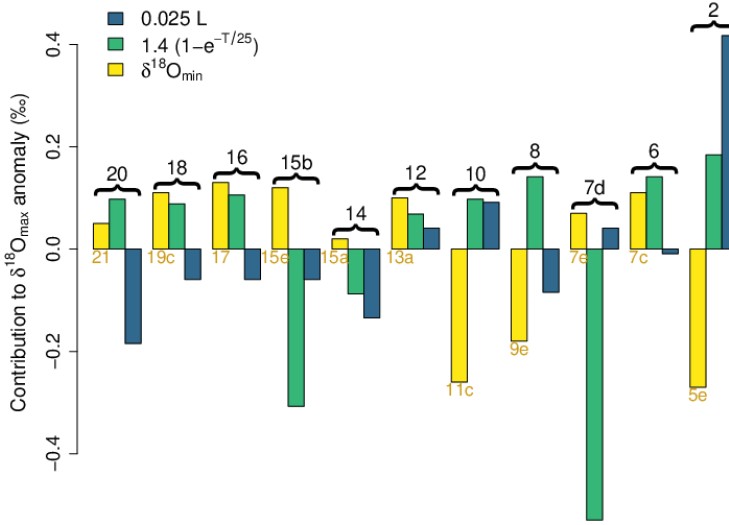

**(b) Model for glacial intensities**


**Figure 6: (a) Contributions of different factors to the predicted $\delta^{18}O_{min}$ anomaly estimated by model 3 using $\delta^{18}O_{max}$ and $I_{AV}$. The height of each bar indicates the anomaly of each term from its average over 11 interglacials. The labels on the top of bars show corresponding Marine Isotope Stages (MIS) for $\delta^{18}O_{min}$. The labels below bars show those for $\delta^{18}O_{max}$. (b) Contributions of different factors to the predicted $\delta^{18}O_{max}$ anomaly: the previous interglacial peak value $\delta^{18}O_{min}$ in the actual data (yellow), the**
**contribution of the basic time dependence, $1.4(1 - e^{-t/25})$ (green), and the contribution of the total time of the low insolation spells during the glacial period, $0.025L$ (dark blue). The height of each bar indicates the anomaly of each term from its average over 11 glacials. The labels on the top of bars show corresponding MISs for $\delta^{18}O_{max}$. The labels below bars show those for $\delta^{18}O_{min}$.**

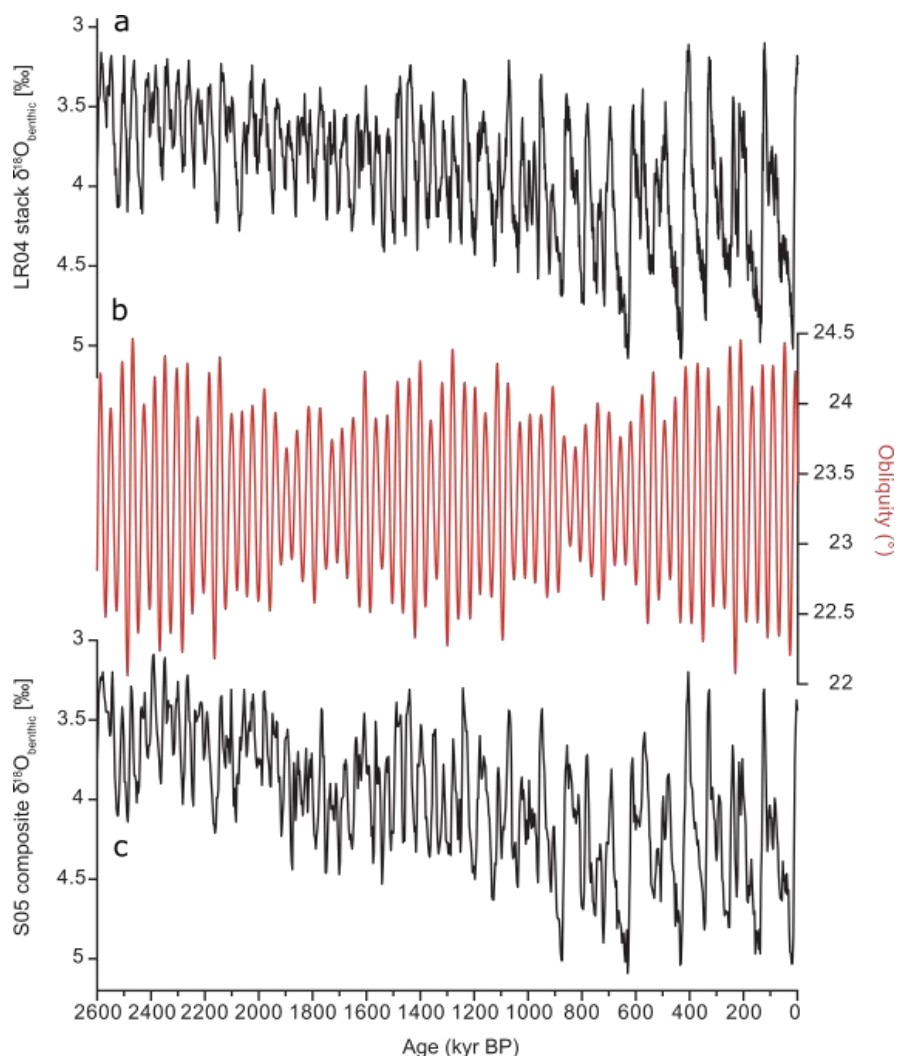

**Figure 7: (a) LR04 benthic $\delta^{18}$O stack (Lisiecki and Raymo, 2005). (b) Obliquity (Laskar et al., 2004). (c) Shackleton05 composite benthic $\delta^{18}$O record from the Eastern Equatorial Pacific (Tzedakis et al., 2017).**


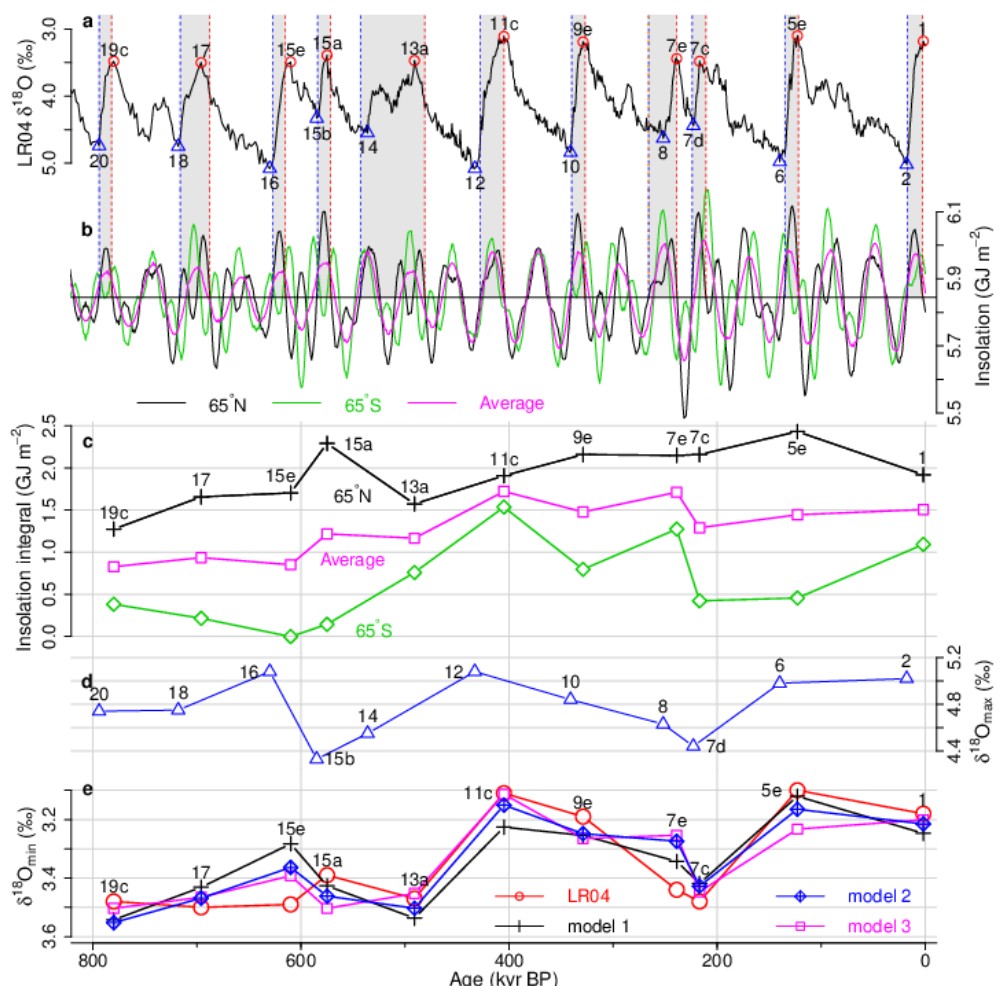

**Figure A1: Same as Fig. 2 but without the exception for the start timing of the Termination III leading to MIS7e. The final correlation with the observed $\delta^{18}O_{min}$ (red circle) decreases from 0.83 to 0.81 in model 1, from 0.94 to 0.86 in model 2, and from 0.92 to 0.83 in model 3 (see also Table A1). However the main features of interglacials intensities are reproduced in models 2 and 3. (a) LR04 $\delta^{18}O$. The red circles indicate the minima of $\delta^{18}O$ ($\delta^{18}O_{min}$) at each interglacial, and the blue triangles the maxima ($\delta^{18}O_{max}$) at glacials. See below for the grey strips and the dashed lines. (b) Caloric summer half-year insolation at 65°N ($F_N$, black) and 65°S ($F_S$, green). The average of the two (magenta) is also shown. The blue dashed lines show timings $t_s$ at which the caloric summer half-year insolation at 65°N exceeds average $5.845$ GJ m$^{-2}$ (black horizontal line) and the red dashed lines show timings $t_e$ at which the insolation falls back below the average. Each termination starts roughly around $t_s$, and it is completed around $t_e$. The grey strips show the termination intervals $[t_s, t_e]$ based on the insolation curve. (c) Integral of the caloric summer insolation anomaly between $t_s$ and $t_e$ at 65°N, $I_N$ (black cross), the integral at 65°S for the same period, $I_S$ (green diamond), and the average $I_{AV} = \frac{1}{2}(I_N + I_S)$ (magenta square). (d) $\delta^{18}O_{max}$. (e) Predictions by linear regression models with explanatory variables in (c) and (d): Model 1 with $I_N$ (black cross); model 2 with both $I_N$ and $I_S$ with their own coefficients (blue diamond with cross); model 3 with $I_{AV}$ (magenta squares).**

| | $\beta_0$ | $\beta_1$ | $\beta_2$ | $\beta_3$ | $p$ | $R$ (correlation) | $R^2$ | BIC |
|---|---|---|---|---|---|---|---|---|
| **Model 1** | 5.74*** | -0.385* | -0.294* | $\equiv 0$ | <0.01 | 0.83 | 0.70 | -13.0 |
| **Model 2** | 5.34*** | -0.283* | -0.289** | -0.170** | <0.001 | 0.94 | 0.89 | -21.9 |
| **Model 3** | 5.07*** | -0.248* | -0.434*** | $\equiv 0$ | <0.001 | 0.92 | 0.85 | -21.1 |

Table 1: Coefficients and statistics of the regression models for interglacial intensity $\delta^{18}O_{min}$ (corresponding to Fig. 2). Model 1 ( $\delta^{18}O_{min}=\beta_0 + \beta_1\delta^{18}O_{max} + \beta_2 I_N$ ), Model 2 ( $\delta^{18}O_{min}=\beta_0 + \beta_1\delta^{18}O_{max} + \beta_2 I_N + \beta_3 I_S$ ), and Model 3 ( $\delta^{18}O_{min}=\beta_0 + \beta_1\delta^{18}O_{max} + \beta_2 I_{AV}$ ). The overall F-test provides a p-value less than 0.01 in each model, which rejects the null hypothesis that none of the variables in the model are significant. The asterisks indicate the significance of each coefficient: * for $p \in (0.01, 0.05]$, ** for $p \in (0.001, 0.01]$, and *** for $p \in [0, 0.001]$.

| | $\beta_0$ | $\beta_1$ | $\beta_2$ | $p$ | $R$ (correlation) | $R^2$ | BIC |
|---|---|---|---|---|---|---|---|
| **With intercept** | -0.033 | 1.4*** | 0.025** | <0.001 | 0.94 | 0.88 | $-8.32$ |
| **Without intercept** | $\equiv 0$ | 1.4*** | 0.025** | $<10^{-10}$ | 0.94 | 0.87 | $-10.7$ |

Table 2: Coefficients and statistics of the regression models for glacial intensities (corresponding to Fig. 4). The model is given as $\delta^{18}O_{max}-\delta^{18}O_{min} = \beta_0 + \beta_1\left(1 - e^{-\frac{T}{25}}\right) + \beta_2 L$. The overall F-test provides a p-value less than 0.001 in each model, which rejects the null hypothesis that none of the variables in the model are significant. The asterisks indicate the significance of each coefficient: * for $p \in (0.01, 0.05]$, ** for $p \in (0.001, 0.01]$, and *** for $p \in [0, 0.001]$.

| | $\beta_0$ | $\beta_1$ | $\beta_2$ | $\beta_3$ | $p$ | $R$ (correlation) | $R^2$ | BIC |
|---|---|---|---|---|---|---|---|---|
| **Model 1** | 5.84*** | -0.410* | -0.279* | $\equiv 0$ | 0.013 | 0.81 | 0.66 | -11.7 |
| **Model 2** | 5.58*** | -0.351* | -0.254* | -0.104 | 0.016 | 0.86 | 0.75 | -12.5 |
| **Model 3** | 5.21*** | -0.305* | -0.320*** | $\equiv 0$ | 0.009 | 0.83 | 0.69 | -12.8 |

Table A1: Coefficients and statistics of the regression models for the data corresponding to Fig. A1 (without the exception for the start timing of the Termination III leading to MIS7e). Model 1 ($\delta^{18}O_{min}=\beta_0 + \beta_1\delta^{18}O_{max} + \beta_2 I_N$), Model 2 ($\delta^{18}O_{min}=\beta_0 + \beta_1\delta^{18}O_{max} + \beta_2 I_N + \beta_3 I_S$), and Model 3 ($\delta^{18}O_{min}=\beta_0 + \beta_1\delta^{18}O_{max} + \beta_2 I_{AV}$). The overall F-test provides a p-value less than 0.05 in each model, which rejects the null hypothesis that none of the variables in the model are significant. The asterisks indicate the significance of each coefficient: * for $p \in (0.01, 0.05]$, ** for $p \in (0.001, 0.01]$, and *** for $p \in [0, 0.001]$.