# Peer review of "Insolation evolution and ice volume legacies determine interglacial and glacial intensity"

_Climate of the Past, 2022_

## Referee Comment (RC1)

**Review of**
***Insolation evolution and ice volume legacies determine interglacial and glacial intensity***
**by Mitsui, Tzedakis and Wolff**

**General comments**

The authors develop regression models to predict the magnitudes of interglacial and glacials over the last 800ka, as quantified by maxima and minima in benthic $\delta^{18}O$.

They first derive a series of models for interglacial intensity as functions of previous glacial strength $\delta^{18}O_{max}$ and half-year summer caloric insolation at 65°N and 65°S temporally integrated across the termination, $I_S$ and $I_N$. They apply BIC to convincingly conclude that all three of these inputs are necessary and able to explain 89% of the variance in interglacial intensities. A model based on $I_{AV} = (I_N+I_S)/2$ is preferred, and is significantly better than a model based on only $I_N$.

To predict glacial intensity, the authors build a more complex regression model that is a function of i) previous interglacial strength $\delta^{18}O_{min}$, ii) a temporal term that depends upon the length of the glacial and assumes a linear relaxation of $\delta^{18}O$ towards $\delta^{18}O_{min} + \beta_1$ with a timescale of 25kyr, and iii) a second temporal term, being the time during the glacial when caloric summer insolation at 65°N is below an empirical threshold. Although this model explains 86% of the variance in glacial strength, I have some concerns that it is overly complex given the small size of the training dataset (11 data points).

The regression relationships are used to decompose the dependence of interglacial/glacial strengths into the different driving factors. The authors conclude that increased obliquity, which drives the variability in $I_{AV}$, explains the stronger interglacials after 430ka, at which time the insolation term switches from being a negative contribution to a positive contribution to interglacial strength.

The work is interesting, novel and within the scope of CP. The manuscript is clearly written and appropriately referenced. Related work is credited, though the paper would benefit from discussion of related work by the lead author, see below. The data (benthic $\delta^{18}O$, temporally integrated summer insolation at 65°N and 65°S $I_N$ and $I_S$, the duration of the glacial T, and the time during which insolation is below the threshold L) are clearly defined and the sources referenced, and the work is therefore reproducible. However, it would be useful if these post-processed data were included as supplementary material. I for one would have been interested to spend a few hours exploring these data but did not have the time to reproduce them from scratch.

**Specific comments**

Why was the period only after 800ka chosen? The LR04 stack and insolation data extend back far earlier that this, and it seems a potentially missed opportunity for additional training data, at least going back a couple of interglacials to the Mid Pleistocene Revolution?

The is some conceptual overlap with a recent publication by the lead author (Mitsui and Boers, 2021, QSR), which used machine learning to similarly conclude that the MBE can be explained by increased obliquity. Some discussion of the distinctions and what this new paper brings would be useful.

The regressions for interglacial intensities are simple and convincing. However, the regression for glacial intensities would benefit from some additional explanation and sensitivities.

$$\delta^{18}O_{max} - \delta^{18}O_{min} = \beta_0 + \beta_1\left(1 - e^{-T/25}\right) + \beta_2 L$$

This equation contains two hidden parameters, i.e. the 25kyr timescale and the empirical insolation threshold of 5.735 GJ m$^{-2}$. This means we have an equation with five parameters which is being fitted to 11 data points and suggests some risk of overfitting. This potential concern should be discussed.

How was the empirical threshold of 5.735 GJ m$^{-2}$ that is used to calculate L determined? It looks like ~95% confidence to yield a positive d18O gradient, which seems reasonable enough but all the same a little arbitrary? More importantly, how sensitive is the model to this choice and are the conclusions robust with respect to the uncertainty in this value?

The second and third terms both represent a form of time dependency (could the third term in effect be a correction for the uncertainty in $\tau$, which is fixed at 25, but lies between 10 and 50 kyr, perhaps depending upon the period of low insolation?). It would feel more natural (to me at least) to instead have separate terms in time and energy. The authors note that the model is rather insensitive to this choice in Figure S2, so I wonder why they chose the model with 'time below threshold' rather than 'integrated insolation below threshold'?

Related, in the S2 version of the model it's not clear to me that the insolation threshold is necessarily needed. Would a simple integral of the insolation from tmin to tmax generate a useful model? If so, this would eliminate the need for the threshold parameter and would make a simpler and more convincing model.

**Technical corrections**

Line 36, missing "," after $\delta^{18}O$.

Line 186 "between 2 and 4 parameters". I'm not certain whether you are neglecting $\tau$, threshold insolation or $\beta_0$ here. I guess $\beta_0$ as it is not favoured by BIC, but this worth clarifying.

Table 2. $R^2$ of 0.99 for "Without intercept" model looks wrong, it should be ~0.86?

---

## Author Comment (AC1)

**Reply to Referee 1**

First of all, thank you very much for reviewing our manuscript in detail and giving us very useful feedback. In what follows, we respond to your comments and questions, point by point, and propose several changes to the manuscript. We consider that these changes will substantially improve the quality and clarity of our manuscript.

In order to improve the readability of our replies we applied a color/type coding to discriminate our replies from the referee's comments. We have attached our replies as a pdf document since color coding is not available in the browser based text editor.

Color/type coding:
*Comment by the referee.*
Reply from the authors.

*General comments*

*The authors develop regression models to predict the magnitudes of interglacial and glacials over the last 800ka, as quantified by maxima and minima in benthic $\delta^{18}O$. They first derive a series of models for interglacial intensity as functions of previous glacial strength $\delta^{18}O_{max}$ and half-year summer caloric insolation at 65°N and 65°S temporally integrated across the termination, $I_S$ and $I_N$.*

*They apply BIC to convincingly conclude that all three of these inputs are necessary and able to explain 89% of the variance in interglacial intensities. A model based on $I_{AV}$ = ($I_N$+$I_S$)/2 is preferred, and is significantly better than a model based on only $I_N$.*

*To predict glacial intensity, the authors build a more complex regression model that is a function of i) previous interglacial strength $\delta^{18}O_{min}$, ii) a temporal term that depends upon the length of the glacial and assumes a linear relaxation of $\delta^{18}O$ towards $\delta^{18}O_{min} + \beta_1$ with a timescale of 25kyr, and iii) a second temporal term, being the time during the glacial when caloric summer insolation at 65°N is below an empirical threshold. Although this model explains 86% of the variance in glacial strength, I have some concerns that it is overly complex given the small size of the training dataset (11 data points).*

*The regression relationships are used to decompose the dependence of interglacial/glacial strengths into the different driving factors. The authors conclude that increased obliquity, which drives the variability in $I_{AV}$, explains the stronger interglacials after 430ka, at which time the insolation term switches from being a negative contribution to a positive contribution to interglacial strength.*

Thank you for summarizing our results. We address your concern about the complexity of the model for glacial strengths below.

*The work is interesting, novel and within the scope of CP. The manuscript is clearly written and appropriately referenced. Related work is credited, though the paper would benefit from discussion of related work by the lead author, see below. The data (benthic $\delta^{18}O$, temporally integrated summer insolation at 65°N and 65°S $I_N$ and $I_S$, the duration of the glacial T, and the time during which insolation is below the threshold L) are clearly defined and the sources referenced, and the work is therefore reproducible. However, it would be useful if these postprocessed data were included as supplementary material. I for one would have been interested to spend a few hours exploring these data but did not have the time to reproduce them from scratch.*

Thank you for your interest and we apologise that you were not able to carry out the tests because the data were not readily available. We will provide data and some programing codes used in this work as Supplementary data.

*Specific comments*

*Why was the period only after 800ka chosen? The LR04 stack and insolation data extend back far earlier that this, and it seems a potentially missed opportunity for additional training data, at least going back a couple of interglacials to the Mid Pleistocene Revolution?*

Thank you for raising this point. We have focused on the last 800 kyr for two reasons. (1) While there is a broad consensus on which $\delta^{18}O$-peaks correspond to interglacials (and which do not) over the last 800 kyr [Past Interglacials Working Group of PAGES, 2016], there is comparably larger uncertainty as well as debate in the classification before 800 kyr BP [Tzedakis et al. 2017; Köhler et al. 2020]. (2) A single model with the same coefficients would not work throughout the time interval across the Mid-Pleistocene Transition (MPT) (~900 kyr BP [Elderfield et al. 2012] or 1250−700 kyr BP [Clark et al. 2006]). This is not surprising, as it is clear that the data show a change in

the response to similar orbital forcing before and after the MPT. Actually we have conducted some preliminary analyses for the extension of the model across the MPT to see whether the same models, with or without changed parameters, can be used. However, the results are too much to be added in this article and deserve further investigation. Thus, we would like to address this in a future study. We will mention this in the revised article.

*There is some conceptual overlap with a recent publication by the lead author (Mitsui and Boers, 2021, QSR), which used machine learning to similarly conclude that the MBE can be explained by increased obliquity. Some discussion of the distinctions and what this new paper brings would be useful.*

Thank you for pointing out this. We will add the following sentence in the revised manuscript in Discussion (paragraph entitled MBE). "Recently Mitsui and Boers (2021) have constructed an Artificial Neural Network model that performs a short-term prediction of $\delta^{18}O$. Through sensitivity analyses, they concluded that the intensification of interglacials across the MBE is attributed to the amplitude increase in the obliquity forcing. This is in line with the present result. On the other hand, the present linear regression model is more physically interpretable and more precise in predicting $\delta^{18}O_{min}$ than the Neural Network model in Mitsui and Boers (2021)."

*The regressions for interglacial intensities are simple and convincing. However, the regression for glacial intensities would benefit from some additional explanation and sensitivities.*

$$\delta^{18}O_{max} - \delta^{18}O_{min} = \beta_0 + \beta_1 \left( 1 - e^{-\frac{T}{25}} \right) + \beta_2 L.$$

*This equation contains two hidden parameters, i.e. the 25kyr timescale and the empirical insolation threshold of 5.735 GJ m$^{-2}$ . This means we have an equation with five parameters which is being fitted to 11 data points and suggests some risk of overfitting. This potential concern should be discussed.*

Thank you for pointing out this. There are indeed 5 potential parameters in total. However, since the 25-kyr timescale is selected based on a large number of data points (LR04 series) in Fig. 5a, this parameter has a different status from the other 4 parameters. Also in the adopted model, the parameter $\beta_0$ is not used to fit the data. Therefore, really free the parameters involved in fitting 11 data points are three: $\beta_1$, $\beta_2$,

and the threshold 5.735. We will clarify this in the revised manuscript. In machine learning, the risk of overfitting is often assessed by cross-validation. In the revised manuscript, we will show, with the leave-one-out cross validation, that the risk of overfitting is low.

*How was the empirical threshold of 5.735 GJ m$^{-2}$ that is used to calculate L determined? It looks like ~95% confidence to yield a positive d18O gradient, which seems reasonable enough but all the same a little arbitrary? More importantly, how sensitive is the model to this choice and are the conclusions robust with respect to the uncertainty in this value?*

In the revised manuscript, we will provide a supplementary figure for a sensitivity analysis with respect to the threshold value.

*The second and third terms both represent a form of time dependency (could the third term in effect be a correction for the uncertainty in $\tau$, which is fixed at 25, but lies between 10 and 50 kyr, perhaps depending upon the period of low insolation?). It would feel more natural (to me at least) to instead have separate terms in time and energy. The authors note that the model is rather insensitive to this choice in Figure S2, so I wonder why they chose the model with 'time below threshold' rather than 'integrated insolation below threshold'?*

It is because the model in Fig. 4 reproduces the intensities of strong glacials, including MIS12, relatively well, while the slightly different model in Fig. S2 does not reproduce the level of MIS12 well. However, the correlation coefficients of Figs. 4 and S2 are indistinguishable. We mention this point more explicitly.

*Related, in the S2 version of the model it's not clear to me that the insolation threshold is necessarily needed. Would a simple integral of the insolation from tmin to tmax generate a useful model? If so, this would eliminate the need for the threshold parameter and would make a simpler and more convincing model.*

Following this suggestion, we have investigated the model with the simple integral of the insolation from $t_{min}$ to $t_{max}$. However, we couldn't get a result better than or comparable with the models already in the manuscript. Thus we have concluded that a nonlinear response to the lower insolation spells is essential to model the glacial intensity. In that case, introducing a threshold is the simplest way to represent the nonlinearity. We will mention this point in the revised manuscript.

*Line 36, missing ",” after δ18O.*

We add the comma: "benthic $\delta^{18}O$, atmospheric $CO_2$".

*Line 186 "between 2 and 4 parameters”. I'm not certain whether you are neglecting $\tau$, threshold insolation or $\beta_0$ ( here. I guess $\beta_0$ ( as it is not favoured by BIC, but this worth clarifying.*

We will correct this.

*Table 2. $R^2$ of 0.99 for "Without intercept” model looks wrong, it should be ~0.86?*

Yes, it should be 0.86. Thank you for this comment.

---

## Author Comment (AC2)

**Reply to Referee 2**

First of all, thank you very much for reviewing our manuscript in detail and giving us very useful feedback. In what follows, we respond to your comments and questions, point by point, and propose several changes to the manuscript. We consider that these changes will substantially improve the quality and clarity of our manuscript.

In order to improve the readability of our replies we applied a color/type coding to discriminate our replies from the referee's comments. We have attached our replies as a pdf document since color coding is not available in the browser based text editor.

Color/type coding:
*Comment by the referee.*
Reply from the authors.

*Based on existing data sets - the LR04 stack and the caloric summer half year insolation - and multiple linear regression models, this study concludes that (1) the interglacial intensity of the last 800 kyr depends on the strength of the previous glacial and summer insolation at high latitudes in both hemispheres, (2) the MBE can be explained by the larger amplitude of obliquity cycles after 430 kyr, and (3) the glacial intensity depends on the strength of the previous interglacial, the time elapsed from it and the evolution of boreal summer insolation. It provides some interesting ideas in explaining the glacial and interglacial intensity. However, I have several major concerns on the methodology used in this study which prevent me to be convinced of the conclusions. Hope the authors will provide more explanations and clarifications.*

*Specific comments and questions:*

*The authors use the LR04 stack as a reference and define the glacial and interglacial intensity by using the delta 18O max and min. In their linear regression models, summer insolation is involved in predicting the delta 18O max and min. As the LR04 stack is orbitally tuned, I wonder to which extent the results are influenced by circular reasoning and whether the comparison between different data sets in Fig2 makes sense.*

First, we have assumed that the orbital tuning in the LR04 $\delta^{18}O$ stack record is right, at least on orbital time scales. Thus, we take it for granted which insolation peak

induces which interglacial. Under this assumption, we have explored the relationships between the amplitude (not the timing) of $\delta^{18}O$ peaks and the insolation forcing.

Taking first the model for the interglacial intensities, there is no circular reasoning in the calculation of the two insolation integrals $I_N$ and $I_S$, since they are calculated purely from the insolation curve. The values of $\delta^{18}O_{min}$ and $\delta^{18}O_{max}$ can be affected by the way benthic records are averaged and combined in constructing the LR04 stack. However, comparison with the more recent probabilistic stack of Ahn et al. (2017) and with the Shackleton S05 eastern Equatorial Pacific composite record (see Tzedakis et al., 2017) does not show any major deviations in interglacial and glacial amplitudes over the last 800 kyr. Moreover, the model does not rely on the absolute ages and use only isotopic levels. Thus we consider that the effect of orbital tuning is minimal in our model for interglacial intensities.

In the model for the glacial intensities, the absolute ages of $\delta^{18}O_{min}$ and $\delta^{18}O_{max}$ are involved in the model. Thus, the orbital tuning could affect our result, but it is not correct to say that our reasoning is circular, since the amplitude of peaks (which we address here) was not involved in the age tuning.

In the revised manuscript, we will address the fact that we have assumed the orbital tuning, and our results must be tested when a new better age model appears.

*2.    CO2 is an important factor in the climate system, but it is not considered in the regression models in predicting the glacial and interglacial intensity.*

Of course, there is no doubt that $CO_2$ is an important factor in the climate system. However, if we consider that the orbital forcing is the only external driver of the system, then $CO_2$ is a feedback, and cannot be used as a predictor of other climate variables. It would be rather simple and not very novel to predict such variables (including $\delta^{18}O_{min}$ (or $\delta^{18}O_{max}$)) from $CO_2$, but this merely begs the question of what drives $CO_2$ concentrations. Instead, given that the $\delta^{18}O$ (convolved ice volume and deep-water temperature signal) is a robust integrated metric of interglacial and glacial intensities (Past Interglacials Working Group of PAGES, 2016), we try to predict $\delta^{18}O_{min}$ or $\delta^{18}O_{max}$ based solely on the insolation curve. In other words we consider that the effect of $CO_2$ is reflected in the outcomes $\delta^{18}O_{min}$ and $\delta^{18}O_{max}$, and in our discussion we consider the role that $CO_2$ may be playing in the mechanistic link between insolation and $\delta^{18}O_{min}$ or $\delta^{18}O_{max}$.

*3.     There are many assumptions made artificially without clear physical meaning. This makes the study appear more like a mathematical game.*

We admit that our models are mathematical. Nevertheless, we believe that our models elucidate physical elements that would have to be taken into account in explaining or simulating interglacial or glacial intensities.

*For example, what is the physical meaning of averaging the 65N and 65S summer insolation,*

We agree that the average of the summer insolation at 65N and 65S, is itself conceptual. In our model for interglacial intensity ($\delta^{18}O_{\min}$), we consider each of the insolation terms separately, and we later discuss the role that each might play. We find that the best models have rather similar coefficients for each of the two insolation terms, and that a more parsimonious model (with fewer parameters) treats them as having the same coefficient. We then note that the average closely follows obliquity which allows us to discuss our results in terms of this easily understood factor.

*why the threshold value 5.735 GJm-2 is chosen, what is its physical meaning,*

The threshold is a parameter introduced to simply model that $\delta^{18}O$ increases rapidly when the insolation level is low (Fig. 5b). However, the value 5.735 GJm$^{-2}$ is chosen to have a good fit. We will explicitly clarify this point in the revised manuscript.

*what is the reasoning of the assumptions on the relation between delta 18O min and mas (line 174-176; line 132).*

In the model for glacial amplitude we assume that the $\delta^{18}O_{max}$ value depends also on whether there is remaining ice (represented by $\delta^{18}O_{\min}$) in the previous interglacial. In the model for interglacial amplitude, we simply note the observation by previous authors that strong interglacials tend to follow strong glacials, and we find that this is indeed a useful predictor. In the discussion we consider the possible physical basis for this.

*4.     The authors attribute the MBE in the LR04 stack to the amplitude change of obliquity, but the physical mechanism is not clear. Moreover, obliquity has a periodicity of 40 kyr, but the interglacial peaks are separated by ~100 kyr. It is unclear to me how the two could be linked.*

The problem of ~100-kyr periodicity has been addressed in previous works. Our previous study with Michel Crucifix shows that every interglacial appears when the caloric summer half-year insolation at 65N exceeds a threshold that decreases with elapsed time since the precious interglacial onset (Tzedakis et al., 2017). This explains how one or two obliquity cycles are skipped without having terminations. The present study (on how the interglacial/glacial strength is determined) is partly based on the previous work (on when interglacials occur). In the revised manuscript, we will mention the model in the previous work on when interglacials occur (Tzedakis et al., 2017) in order to make clear the linkage to the present study.

With respect to the shift in interglacial intensities at the MBE, we suggest that the this may be attributed to the long 1.2 million year cycle that modulates the amplitude of obliquity (so greater obliquity maxima after 430 ka led to stronger interglacials).

*There is also MBE in the interglacial CO2 concentration. I wonder why CO2 is not mentioned in explaining the MBE.*

In fact, we have mentioned the role of $CO_2$ in Introduction and Discussion (lines 47, 199, 204, 224) citing literature. However as we have discussed above, the whole purpose of the present paper is to predict the amplitude of interglacials and glacials from the insolation, with $CO_2$ as part of the mechanism leading to the observed climate variables (the benthic $\delta^{18}O$). That is, $CO_2$ is considered as an agent in-between insolation and $\delta^{18}O$ change. We will mention this last point in Data and Methods in the revised manuscript.

*5.    Can the regression models based on the last 800kyr data explain the glacial and interglacial intensity before 800kyr?*

Thank you for asking this point. Actually, we have investigated if the same form of the model works also before 800 kyr BP. Our preliminary results are positive. However, it appears that prolongation and intensification of glacial-interglacial cycles across the Mid-Pleistocene Transition (MPT) does not allow the use of the same coefficients in the model across the MPT. This makes the modelling rather complicated. This will be the subject of a future study. We will mention this in the revised article.

*6.    There is no real conclusion section.*

The last section was called Summary and Discussion, rather than Conclusion, because the former name represents the section more suitably. In Summary and Discussion, we have concluding remarks, e.g., the very last paragraph, Lines 241-244. Nevertheless, we make concluding statements clearer in the revised manuscript.

---

## Author Response (AR1)

Dr. Takahito Mitsui

Earth System Modelling, School of Engineering & Design

Technical University of Munich

80333 Munich, Germany

takahito321@gmail.com

31 July 2022

Dear Editor Prof. Erin McClymont,

Thank you very much for reviewing our manuscript entitled "Insolation evolution and ice volume legacies determine interglacial and glacial intensity". We have revised our manuscript responding to the reviewers' comments and questions, point-by-point. Major changes include

- The reason for the choice of the insolation threshold 5.735 GJ m$^{-2}$ as well as the robustness of the result with respect to this parameter was queried by both referees. Since it has too many digits, we have re-done the analysis shown in Fig. 4, Fig. 6b, and Table 2 by using a simpler value 5.7 GJ m$^{-2}$. Consequently Fig. 4, Fig. 6b and Table 2 are renewed, but the results are virtually unchanged.

- We have included a robustness analysis with respect to the threshold parameter (5.7 GJ m$^{-2}$) and the time constant $\tau$ (25 kyr) (see the last paragraph of Section 3.2 and Fig. S3).

- The prediction performance of the models for unseen data is assessed in new Appendix B, in order to show that the models are not severely overfitted.

- We have clarified the difference between the present study and a publication by the first author (at the end of the 4$^{th}$ paragraph of Section 4).

- The postprocessed data is provided as Supplementary materials (Figure2ab.csv, Figure2acde.csv, Figure4acdefg.csv).

- We have changed the color-coordinate in Fig. 3 so that stronger (weaker) interglacials correspond to warmer (colder) color.

- A few references are added as in the marked-up manuscript.

The other changes are described in the replies to Referees in what follows.

We consider that these changes will substantially improve the quality and clarity of our manuscript.

**Point-by-point reply to the reviewers' comments**

In order to improve the readability of our replies we applied a color/type coding to discriminate our replies from the referee's comments.

Color/type coding:
*Comment by the referee (RC1 or RC2).*
Reply from the authors to referees (AC1 or AC2)
**Changes in the revised paper.**

**Reply to Referee 1**

*It would be useful if these postprocessed data were included as supplementary material. I for one would have been interested to spend a few hours exploring these data but did not have the time to reproduce them from scratch.*

Thank you for your interest and we apologise that you were not able to carry out the tests because the data were not readily available. **In the revised manuscript, the postprocessed data used for Figs. 2 and 4 are provided as supplementary materials:**

- **Figure2ab.csv      [caloric summer half-year insolations at 65N and 65S, obliquity and 1-ka sampled LR04 series]**

- **Figure2acde.csv    [model variables and predicted $\delta^{18}O_{min}$]**

- **Figure4acdefg.csv  [model variables and predicted $\delta^{18}O_{max}$]**

**In Code and data availability, we note that "The R-codes used in this study are available from the corresponding author (Takahito Mitsui) upon reasonable request".**

*Specific comments*

*Why was the period only after 800ka chosen? The LR04 stack and insolation data extend back far earlier that this, and it seems a potentially missed opportunity for additional training data, at least going back a couple of interglacials to the Mid Pleistocene Revolution?*

Thank you for raising this point. We have focused on the last 800 kyr for two reasons. (1) While there is a broad consensus on which $\delta^{18}$O-peaks correspond to interglacials (and which do not) over the last 800 kyr [Past Interglacials Working Group of PAGES, 2016], there is comparably larger uncertainty as well as debate in the classification before 800 kyr BP [Tzedakis et al. 2017; Köhler et al. 2020]. (2) A single model with the same coefficients would not work throughout the time interval across the Mid-Pleistocene Transition (MPT) (~900 kyr BP [Elderfield et al. 2012] or $1250-700$ kyr BP [Clark et al. 2006]). This is not surprising, as it is clear that the data show a change in the response to similar orbital forcing before and after the MPT. Actually we have conducted some preliminary analyses for the extension of the model across the MPT to see whether the same models, with or without changed parameters, can be used. However, the results are too much to be added in this article and deserve further investigation. Thus, we would like to address this in a future study. **In the revised article, we have added the following sentence in the introduction: "In this work, we focus on the interglacial and glacial intensities over the last 800 kyr, where there is a broad consensus over which $\delta^{18}$O-peaks correspond to interglacials (Past Interglacials Working Group of PAGES, 2016). The extension of our models beyond 800 kyr BP will be the subject of a future study."**

*There is some conceptual overlap with a recent publication by the lead author (Mitsui and Boers, 2021, QSR), which used machine learning to similarly conclude that the MBE can be explained by increased obliquity. Some discussion of the distinctions and what this new paper brings would be useful.*

Thank you for pointing out this. **In the revised manuscript, we have add the following (paragraph with heading "MBE" in Section 4): "Recently Mitsui and Boers (2021) have developed an Artificial Neural Network (ANN) model that performs a skilful 21-kyr ahead prediction of $\delta^{18}$O on the basis of the past $\delta^{18}$O history and the insolation evolution. Through the sensitivity analysis of the ANN model, they concluded that the intensification of interglacials across the MBE is attributed to the amplitude increase in the obliquity forcing. While this is consistent with our conclusions, our present regression model is more**

**physically interpretable than the ANN model and even more precise in predicting $\delta^{18}O_{min}$."**

*The regressions for interglacial intensities are simple and convincing. However, the regression for glacial intensities would benefit from some additional explanation and sensitivities.*

$$\delta^{18}O_{max} - \delta^{18}O_{min} = \beta_0 + \beta_1\left(1 - e^{-\frac{T}{25}}\right) + \beta_2 L.$$

*This equation contains two hidden parameters, i.e. the 25kyr timescale and the empirical insolation threshold of 5.735 GJ m$^{-2}$ . This means we have an equation with five parameters which is being fitted to 11 data points and suggests some risk of overfitting. This potential concern should be discussed.*

Thank you for pointing out this. There are indeed maximum 5 parameters in total. However, since the 25-kyr timescale is selected based on a large number of data points (LR04 series) in Fig. 5a, this parameter has a different status from the other 4 parameters. Also in the adopted model, the parameter $\beta_0$ is not used to fit the data. Therefore, really the free parameters involved in fitting 11 data points are three: $\beta_1$, $\beta_2$, and the threshold 5.735. In machine learning, the risk of overfitting is often assessed by cross-validation. **In the revised manuscript, we have shown in Appendix B, with the leave-one-out cross validation method, that the models actually have a prediction ability for unseen data. Thus they are not severely overfitted.**

*How was the empirical threshold of 5.735 GJ m$^{-2}$ that is used to calculate L determined? It looks like ~95% confidence to yield a positive d18O gradient, which seems reasonable enough but all the same a little arbitrary? More importantly, how sensitive is the model to this choice and are the conclusions robust with respect to the uncertainty in this value?*

The empirical threshold 5.735 GJ m$^{-2}$ is chosen because it gives a good prediction in the previous Fig. 4g and also because the time derivative of $\delta^{18}O$ change is high below ~ 5.7 GJ m$^{-2}$ (Fig. 5b). However, too many digits in 5.735 might misleadingly suggest that the result is not robust against changes in this parameter. **Therefore in the revised manuscript:**

- **we adopt the threshold parameter 5.7 GJ m$^{-2}$ instead of 5.735 GJ m$^{-2}$. This hardly changes the result.**

- **we add the result of a sensitivity analysis (Figure S3). It shows that the model prediction is relatively insensitive to the threshold parameter and the time scale constant $\tau$.**

*The second and third terms both represent a form of time dependency (could the third term in effect be a correction for the uncertainty in $\tau$, which is fixed at 25, but lies between 10 and 50 kyr, perhaps depending upon the period of low insolation?). It would feel more natural (to me at least) to instead have separate terms in time and energy. The authors note that the model is rather insensitive to this choice in Figure S2, so I wonder why they chose the model with 'time below threshold' rather than 'integrated insolation below threshold'?*

We adopt the model using the total time of low insolation period as the main result since the fit of the model R=0.90 is slightly better than that of the alternative using the insolation integral (R=0.89). **This was mentioned at the end of the 3$^{rd}$ paragraph of Section 3.2.**

*Related, in the S2 version of the model it's not clear to me that the insolation threshold is necessarily needed. Would a simple integral of the insolation from tmin to tmax generate a useful model? If so, this would eliminate the need for the threshold parameter and would make a simpler and more convincing model.*

Following this suggestion, we have investigated the model with the simple integral of the insolation from $t_{min}$ to $t_{max}$. However, we couldn't get a result better than or comparable with the models already in the manuscript. Thus we have concluded that a nonlinear response to the lower insolation spells is essential to model the glacial intensity. In that case, introducing a threshold is the simplest way to represent the nonlinearity. **In the revised manuscript we mention**

- **"In either case, a deficit of insolation is a good predictor of glacial intensities. Thresholding is the simplest way to capture insolation deficits" (at the end of the 3$^{rd}$ paragraph of Section 3.2)**
- **"Another model using the simple integral $\sum F_N(t)$ without thresholding gives a lower prediction skill R=0.78" (caption of Fig. S2)**

*Line 36, missing "," after δ18O.*

We add the comma: "benthic $\delta^{18}O$, atmospheric $CO_2$".

*Line 186 "between 2 and 4 parameters". I'm not certain whether you are neglecting $\tau$, threshold insolation or $\beta_0$ ( here. I guess $\beta_0$ ( as it is not favoured by BIC, but this worth clarifying.*

**We removed the corresponding part and added a sentence that clarifies this in the last paragraph of Section 3.2: "The final model has four parameters: regression coefficients $\beta_{1,2}$, time constant $\tau\,(= 25\,\text{kyr})$ and the threshold defining the low-insolation period (5.7 GJ m$^{-2}$)."**

*Table 2. $R^2$ of 0.99 for "Without intercept" model looks wrong, it should be ~0.86?*

**We corrected it from 0.99 to 0.87.** Thank you for this comment.

**Reply to Referee 2**

*The authors use the LR04 stack as a reference and define the glacial and interglacial intensity by using the delta 18O max and min. In their linear regression models, summer insolation is involved in predicting the delta 18O max and min. As the LR04 stack is orbitally tuned, I wonder to which extent the results are influenced by circular reasoning and whether the comparison between different data sets in Fig2 makes sense.*

First, we have assumed that the orbital tuning in the LR04 $\delta^{18}O$ stack record is right, at least on orbital time scales. Thus, we take it for granted which insolation peak induces which interglacial. Under this assumption, we have explored the relationships between the amplitude (not the timing) of $\delta^{18}O$ peaks and the insolation forcing.

Taking first the model for the interglacial intensities, there is no circular reasoning in the calculation of the two insolation integrals $I_N$ and $I_S$, since they are calculated purely from the insolation curve. The values of $\delta^{18}O_{min}$ and $\delta^{18}O_{max}$ can be affected by the way benthic records are averaged and combined in constructing the LR04 stack. However, comparison with the more recent probabilistic stack of Ahn et al. (2017) and with the Shackleton S05 eastern Equatorial Pacific composite record (see Tzedakis et al., 2017) does not show any major deviations in interglacial and glacial amplitudes

over the last 800 kyr. Moreover, the model does not rely on the absolute ages and use only isotopic levels. Thus we consider that the effect of orbital tuning is minimal in our model for interglacial intensities.

In the model for the glacial intensities, the absolute ages of $\delta^{18}O_{min}$ and $\delta^{18}O_{max}$ are involved in the model. Thus, the orbital tuning could affect our result, but it is not correct to say that our reasoning is circular, since the amplitude of peaks (which we address here) was not involved in the age tuning.

**In the revised manuscript, we have addressed that we have assumed the orbital tuning in LR04 record (in the 1ˢᵗ paragraph in Method and Data as well as at the end of Section 3.2).**

*2.    CO2 is an important factor in the climate system, but it is not considered in the regression models in predicting the glacial and interglacial intensity.*

Of course, there is no doubt that $CO_2$ is an important factor in the climate system. However, if we consider that the orbital forcing is the only external driver of the system, then $CO_2$ is a feedback, and cannot be used as a predictor of other climate variables. It would be rather simple and not very novel to predict such variables (including $\delta^{18}O_{min}$ (or $\delta^{18}O_{max}$)) from $CO_2$, but this merely begs the question of what drives $CO_2$ concentrations. Instead, given that the $\delta^{18}O$ (convolved ice volume and deep-water temperature signal) is a robust integrated metric of interglacial and glacial intensities (Past Interglacials Working Group of PAGES, 2016), we try to predict $\delta^{18}O_{min}$ or $\delta^{18}O_{max}$ based solely on the insolation curve. In other words we consider that the effect of $CO_2$ is reflected in the outcomes $\delta^{18}O_{min}$ and $\delta^{18}O_{max}$, and in our discussion we consider the role that $CO_2$ may be playing in the mechanistic link between insolation and $\delta^{18}O_{min}$ or $\delta^{18}O_{max}$.

**At the end of Introduction, we added a statement: "Thus, the whole purpose of the present paper is to predict the amplitude of $\delta^{18}O$ from the insolation, the only external driver of the climate system. Atmospheric CO₂ or other climate feedbacks are considered as agents in-between insolation and $\delta^{18}O$ changes and their potential role is discussed in the final section."**

*3.    There are many assumptions made artificially without clear physical meaning. This makes the study appear more like a mathematical game.*

We admit that our models are mathematical. Nevertheless, we believe that our models elucidate physical elements that would have to be taken into account in explaining or simulating interglacial or glacial intensities.

*For example, what is the physical meaning of averaging the 65N and 65S summer insolation,*

We agree that the average of the summer insolation at 65N and 65S, is itself conceptual. In our model for interglacial intensity ($\delta^{18}O_{\min}$), we consider each of the insolation terms separately, and we later discuss the role that each might play. We find that the best models have rather similar coefficients for each of the two insolation terms, and that a more parsimonious model (with fewer parameters) treats them as having the same coefficient. We then note that the average closely follows obliquity which allows us to discuss our results in terms of this easily understood factor.

*why the threshold value 5.735 GJm-2 is chosen, what is its physical meaning,*

The threshold value around 5.7 GJ $m^{-2}$ is motivated by the fact that $\delta^{18}O$ almost exclusively grows (at ~10-kyr time scales) for the insolation below ~5.7—5.8 GJ $m^{-2}$. In the revised manuscript we have mentioned this. But at the same time, the specific value was adopted because it gives a good prediction. **In the revised manuscript, we have changed the value from 5.735 to 5.7 GJ $m^{-2}$. The result is virtually unchanged. We also added a result of sensitivity analysis in Fig. S3 (referred at the last paragraph of Section 3.2), which shows that the model prediction is relatively insensitive to the threshold parameter as well as the time scale constant $\tau$.**

*what is the reasoning of the assumptions on the relation between delta 18O min and mas (line 174-176; line 132).*

In the model for glacial amplitude we assume that the $\delta^{18}O_{max}$ value depends also on whether there is remaining ice (represented by $\delta^{18}O_{\min}$) in the previous interglacial. In the model for interglacial amplitude, we simply note the observation by previous authors that strong interglacials tend to follow strong glacials, and we find that this is indeed a useful predictor. In the discussion we consider the possible physical basis for this.

*4.    The authors attribute the MBE in the LR04 stack to the amplitude change of obliquity, but the physical mechanism is not clear. Moreover, obliquity has a periodicity of 40 kyr, but the interglacial peaks are separated by ~100 kyr. It is unclear to me how the two could be linked.*

The problem of ~100-kyr periodicity has been addressed in previous works. Our previous study with Michel Crucifix shows that every interglacial appears when the caloric summer half-year insolation at 65N exceeds a threshold that decreases with elapsed time since the precious interglacial onset (Tzedakis et al., 2017). This explains how one or two obliquity cycles are skipped without having terminations. The present study (on how the interglacial/glacial strength is determined) is partly based on the previous work (on when interglacials occur).

**In order to make our present focus clear, we added the following sentence in the end of the first paragraph of Section 2: "we explore the relationships between the amplitude (not the timing) of $\delta^{18}O$ peaks and the insolation forcing (see Tzedakis et al. 2017 for the timing, which explains how one or two obliquity cycles are skipped without having terminations)."**

With respect to the shift in interglacial intensities at the MBE, we suggest that the this may be attributed to the long 1.2 million year cycle that modulates the amplitude of obliquity (so greater obliquity maxima after 430 ka led to stronger interglacials).

*There is also MBE in the interglacial CO2 concentration. I wonder why CO2 is not mentioned in explaining the MBE.*

In fact, we have mentioned the role of $CO_2$ in Introduction and Discussion (lines 47, 199, 204, 224) citing literature. However as we have discussed above, the whole purpose of the present paper is to predict the amplitude of interglacials and glacials from the insolation, with $CO_2$ as part of the mechanism leading to the observed climate variables (the benthic $\delta^{18}O$). That is, $CO_2$ is considered as an agent in-between insolation and $\delta^{18}O$ change. **In the revised manuscript (in the end of Introduction) we have mentioned: "Thus the whole purpose of the present paper is to predict the amplitude of $\delta^{18}O$ from the insolation. Atmospheric $CO_2$ or other climatic elements are considered as agents in-between insolation and $\delta^{18}O$ change."**

*5.    Can the regression models based on the last 800kyr data explain the glacial and interglacial intensity before 800kyr?*

Thank you for asking this point. Actually, we have investigated if the same form of the model works also before 800 kyr BP. Our preliminary results are positive. However, it appears that prolongation and intensification of glacial-interglacial cycles across the Mid-Pleistocene Transition (MPT) does not allow the use of the same coefficients in the model across the MPT. This makes the modelling rather complicated. This will be the subject of a future study. **In the revised article, we have added the following sentence in the introduction: "In this work, we focus on the interglacial and glacial intensities over the last 800 kyr, where there is a broad consensus over which $\delta^{18}$O-peaks correspond to interglacials (Past Interglacials Working Group of PAGES, 2016). The extension of our models beyond 800 kyr BP will be the subject of a future study."**

*6.    There is no real conclusion section.*

The last section was called Summary and Discussion, rather than Conclusion, because the former name represents the section more suitably. In Summary and Discussion, we had concluding remarks, e.g., the very last paragraph, Lines 241-244. **To make the summary (or conclusion) more clear, we extended the first paragraph of Summary and Discussion in the revised manuscript. Actually we added the following sentence: "While the models contain three to four parameters, they are still simple explanatory frameworks. These models show that interglacial intensity over the last 800-kyr can be described as a function of the strength of the previous glacial and the summer insolation at high latitudes in both hemispheres during the deglaciation, and also glacial intensity is linked with the strength of the previous interglacial, the time elapsed from it, and the evolution of boreal summer insolation."**